# OCEAN SWELL WITHIN THE KINETIC EQUATION FOR WATER WAVES

Sergei I. Badulin[1,2] and Vladimir E. Zakharov[1,2,3,4,5]

[1]P. P. Shirshov Institute of Oceanology of the Russian Academy of Science, Russia
[2]Novosibirsk State University, Russia
[3]University of Arizona, Tuscon, USA
[4]P.N. Lebedev Physical Institute of Russian Academy of Sciences
[5]Waves and Solitons LLC, Phoenix, Arizona, USA

*Correspondence to:* S. I. Badulin (badulin.si@ocean.ru)

**Abstract.** Results of extensive simulations of swell evolution within the duration-limited setup for the kinetic Hasselmann equation for long durations of up to $2 \cdot 10^6$ seconds are presented. Basic solutions of the theory of weak turbulence, the so-called Kolmogorov-Zakharov solutions, are shown to be relevant to the results of the simulations. Features of self-similarity of wave spectra are detailed and their impact on methods of ocean swell monitoring are discussed. Essential drop of wave energy (wave height) due to wave-wave interactions is found at initial stages of swell evolution (of order of $1000$ km for typical parameters of the ocean swell). At longer times wave-wave interactions are responsible for a universal angular distribution of wave spectra in a wide range of initial conditions. Weak power-law attenuation of swell within the Hasselmann equation is not consistent with results of ocean swell tracking from satellite altimetry and SAR (Synthetic Aperture Radar) data. At the same time, the relatively fast weakening of wave-wave interactions makes the swell evolution sensitive to other effects. In particular, as shown, coupling with locally generated wind waves can force the swell to grow at relatively light winds.

## 1  Physical models of ocean swell

Ocean swell is an important constituent of the field of surface gravity waves in the sea and, more generally, of the sea environment as a whole. Swell is usually defined as a fraction of wave field that does not depend (or depends slightly) on local wind. Being generated in confined stormy areas these waves can propagate long distances of many thousand miles, thus, influencing vast ocean stretches. For example, swell from the Roaring Forties in the Southern Ocean can traverse the Pacifica and reach distant shores of California and Kamchatka. Predicting swell as a part of sea wave forecast remains a burning problem for maritime safety and marine engineering.

Pioneering works by Barber and Ursell (1948); Munk et al. (1963); Snodgrass et al. (1966) discovered a rich physics of the phenomenon and gave first examples of accurate measurements of magnitudes, periods and directional spreading of swell. All the articles contain thorough discussions of physical background of swell generation, attenuation and interaction with other

types of ocean motions. A fascinating story of a grand experiment on ocean swell has been presented to a wide audience in the documentary 'Waves across the Pacific' (can be found at https://www.youtube.com/watch?v=MX5cKoOm6Pk).[1]

Nonlinear wave-wave interactions have been sketched by Snodgrass et al. (1966) as a novelty introduced by the milestone papers by Phillips (1960) and Hasselmann (1962). A possible important role of these interactions at high swells for relatively short time of evolution has been outlined and evaluated. The first estimates of the observed rates of swell attenuation have been carried out by Snodgrass et al. (1966) based on observation at near-shore stations. Their e-folding scale about $4000 \, \text{km}$ (distance in which an exponentially decaying wave height decreases by a factor of $e$) is consistent with some of today's results of the satellite tracking of swell (Ardhuin et al., 2009, 2010; Jiang et al., 2016) and with treatment of these results within the model of swell attenuation due to coupling with turbulent atmospheric layer (e.g. Tsimring, 1986; Kantha, 2006). Alternative semi-empirical model of Babanin (2006) predicts quite different algebraic law and stronger swell attenuation at shorter distances from the swell source (Young et al., 2013). Note that the effect of the decay of a monochromatic wave due to turbulent wave flow is found to be quadratic in wave amplitude, i.e. to be of lower-order nonlinearity than in the non-dissipative theory of weakly nonlinear water waves.

It should be stressed that a number of theoretical and numerical models including those mentioned above treats swell as a quasi-monochromatic wave and, thus, ignores nonlinear interactions of the swell harmonics themselves and the swell coupling with locally generated wind waves. The latter effect can be essential as observations and simulations clearly show (e.g. Kahma and Pettersson, 1994; Pettersson, 2004; Young, 2006; Badulin et al., 2008b, and refs. therein). Usually the swell is continued to be considered as a superposition of harmonics that do not interact with each other and, thus, can be described by the well-known methods of the linear theory of waves (e.g. Ewans, 1998; Ewans et al., 2004). Many features of the observed swell can be related to such models. For example, the observed effect of linear growth of the swell frequency in a site can be explained as an effect of dispersion of a linear wave packet over long time and successfully used for relating these observations with stormy areas that generate the swell (e.g. Barber and Ursell, 1948; Ewans et al., 2004).

Synthetic aperture radars (SAR) allow for spatial resolution up to tens of meters (e.g. Ardhuin et al., 2010; Young et al., 2013). Satellite altimeters measure wave height averaged over a snapshot of a few square kilometers. These snapshots are adequate for currently known methods of statistical description of waves in research and application models. These can be used for swell tracking in combination with other tools (e.g. wave models as in Jiang et al., 2016). Re-tracking of swells allows, first, to relate the swell events with their probable sources – stormy areas and, secondly, the swell transformation enables to estimate effects of other motions of the atmosphere and ocean – seasonal wind activity (e.g. Chen et al., 2002), wave-current interaction (e.g. Beal et al., 1997) and bathymetry effects (Young et al., 2013) etc.. Such a work requires adequate physical models of swell propagation and transformation. This paper is aimed to narrow the gap.

Meanwhile, the linear treatment remains quite restrictive and cannot explain important features of swell. The observed swell spectra exhibit frequency downshift which is not predicted by deterministic linear or weakly nonlinear models of narrow-banded wave guide evolution (e.g. data of Snodgrass et al., 1966, and comments on these data by Henderson and Segur (2013)). Moreover, these spectra show invariance of their shapes that is unlikely to appear in linear dispersive wave system. These noted

---

[1]The authors are thankful to Dr. Gerbrant van Vledder, Delft University of Technology for this reference

features are common for wave spectra described by the kinetic equation for water waves, the so-called Hasselmann (1962) equation.

In this paper we present results of extensive simulations of ocean swell within the Hasselmann equation for deep water waves. The simplest duration-limited setup has been chosen to obtain numerical solutions for the duration up to $2 \cdot 10^6$ seconds (about 23 days) for typical parameters of ocean swell (wavelengths $150-400$ meters, wave periods $10-16$ s, initial significant heights $3-15$ meters).

We analyze the simulation results within the framework of the theory of weak turbulence (Zakharov et al., 1992). The slowly evolving swell solutions appear to be quite close to the stationary Kolmogorov-Zakharov spectra. We give a short theoretical introduction and present estimates of the basic constants of the theory in the next section. In sect.3 we relate results of simulations with properties of the self-similar solutions of the kinetic equation. Zaslavskii (2000) was the first to present the self-similar solutions for swell assuming the angular narrowness of the swell spectra and stated explicit analytical results. In fact, more general consideration, in the spirit of Badulin et al. (2002, 2005a), leads to important findings and raises questions independent of the assumption of angular narrowness.

We demonstrate the fact that is usually ignored: the power-law swell attenuation within the conservative kinetic equation. We show that it does not contradict observations mentioned above. We also reveal a remarkable feature of collapsing the swell spectra onto an angular distribution that depends weakly on initial angular spreading. Such universality can be of great value for modelling swell and developing methods for its monitoring (Delpey et al., 2010).

We conclude this paper with a discussion of how to apply this model. Evidently, the setup of duration-limited evolution is quite restrictive and does not reflect essential features of ocean swell when wave dispersion and spatial divergence play a key role. At the same time, wave-wave interactions remain of importance independently of the setup. The weakening of swell evolution is not directly related to abatement of wave-wave interactions which are able to effectively restore perturbations of these quasi-stationary states (Zakharov and Badulin, 2011). On the contrary, this favors coupling of the quasi-stationary swell with ocean environment. In particular, the locally generated wind-driven waves can switch the swell attenuation to swell amplification. This effect can be considered for interpretation of recent observations of swell from space ('negative' dissipation in words of Jiang et al., 2016). Many problems of adequate physical description of swell in the ocean are still open. This paper is an attempt to reveal essential features of swell evolution within the simplest model of the kinetic Hasselmann equation.

## 2 Solutions for ocean swell

### 2.1 The Kolmogorov-Zakharov solutions

In this section we reproduce previously reported theoretical results on evolution of swell as a random field of weakly interacting wave harmonics. We apply the statistical theory of wind-driven seas (Zakharov, 1999) to the sea swell, whose description with this approach, is usually considered questionable. A random wave field is described by the kinetic equation derived by Klauss

Hasselmann (1962) for weakly nonlinear deep water waves in the absence of dissipation and external forcing

$$\frac{\partial N_{\mathbf{k}}}{\partial t} + \nabla_{\mathbf{k}}\omega_{\mathbf{k}}\nabla_{\mathbf{r}}N_{\mathbf{k}} = S_{nl}. \tag{1}$$

Equation (1) is written for the spectral density of wave action $N(\mathbf{k},\mathbf{x},t) = E(\mathbf{k},\mathbf{x},t)/\omega(\mathbf{k})$ ($E(\mathbf{k},\mathbf{x},t)$ is the wave energy spectrum and the wave frequency obeys linear dispersion relation $\omega = \sqrt{g|\mathbf{k}|}$). Subscripts for $\nabla$ corresponds to the two-dimensional gradient operator in the corresponding space of coordinates $\mathbf{x}$ and wavevectors $\mathbf{k}$ (i.e. $\nabla_{\mathbf{r}} = (\partial/\partial x, \partial/\partial y)$).

The right-hand term $S_{nl}$ describes the effect of wave-wave resonant interactions and can be written in explicit form (see Appendices in Badulin et al., 2005a, for collection of formulas). The cumbersome term $S_{nl}$ causes many problems for wave modelling whenever (1) is extensively used. Nevertheless, for the deep water case, one has a key property of homogeneity

$$S_{nl}[\kappa\mathbf{k}, \upsilon N_{\mathbf{k}}] = \kappa^{19/2}\upsilon^3 S_{nl}[\mathbf{k}, N_{\mathbf{k}}]. \tag{2}$$

that helps in acquiring important analytical results. Stretching in $\kappa$ times in wave scale or in $\upsilon$ times in wave action, where $\kappa, \upsilon$ are positive leads to simple re-scaling of the collision term, $S_{nl}$. This important property gives a clue for constructing power-law stationary solutions of the kinetic equation, i.e. solutions for the equation

$$S_{nl} = 0. \tag{3}$$

Two isotropic stationary solutions of (3) correspond to constant fluxes of wave energy and action in wave scales. The direct cascade solution (Zakharov and Filonenko, 1966) in terms of frequency spectrum of energy

$$E^{(1)}(\omega, \theta) = 2C_p\frac{P^{1/3}g^{4/3}}{\omega^4} \tag{4}$$

introduces the basic Kolmogorov constant $C_p$ and describes the energy transfer to infinitely short waves with constant flux $P$. The wave action transfer in the opposite direction of long waves is described by the inverse cascade solution (Zakharov and Zaslavsky, 1982) with wave action flux $Q$ and another Kolmogorov's constant $C_q$:

$$E^{(2)}(\omega, \theta) = 2C_q\frac{Q^{1/3}g^{4/3}}{\omega^{11/3}}. \tag{5}$$

Note, that key features of the isotropic Kolmogorov-Zakharov solutions (4,5) are reproduced quite well by means of direct numerical simulations (DNS) based on the integrodifferential Zakharov equation (Annenkov and Shrira, 2006) or on the primitive Euler equations (Onorato et al., 2002).

An approximate weakly anisotropic Kolmogorov-Zakharov solution has been obtained by Katz and Kontorovich (1974) as an extension of (4)

$$E^{(3)}(\omega, \theta) = 2\frac{P^{1/3}g^{4/3}}{\omega^4}\left(C_p + C_m\frac{gM}{\omega P}\cos\theta + \dots\right). \tag{6}$$

It associates the wave spectrum anisotropy with the constant spectral flux of wave momentum $M$ and the so-called second Kolmogorov constant $C_m$. As it is seen from (6) the solution anisotropy vanishes as $\omega \to \infty$: wave spectra become isotropic

for short waves. The whole set of the KZ solutions (4–6) can be treated naturally within the dimensional approach: these are just particular cases of solutions of the form

$$E^{(KZ)}(\omega) = \frac{P^{1/3} g^{4/3}}{\omega^4} G(\omega Q/P, gM/(\omega P), \theta) \tag{7}$$

where $G$ is a function of dimensionless arguments scaled by spectral fluxes of wave energy $P$, action $Q$ and momentum $M$.

Originally, solutions (4–6) were derived in particularly sophisticated and cumbersome ways. Later on, simpler and more physically transparent approaches have been presented (Zakharov and Pushkarev, 1999; Balk, 2000; Pushkarev et al., 2003, 2004; Badulin et al., 2005a; Zakharov, 2010). These more general approaches allow to find higher-order terms of the anisotropic Kolmogorov-Zakharov solutions (6). In particular, they predict the next term to be proportional to $\cos 2\theta/\omega^2$ which is the second angular harmonics of the stationary solution (6).

Swell solutions evolve slowly with time and, thus, give a good opportunity for discussing features of the KZ solutions (or, alternatively, the KZ solutions can be used as a reference case for the swell studies). One of the key points of this discussion is the question of uniqueness, universality of the swell solutions that can be treated in the context of general KZ solutions (7). The principal terms of the general Kolomogorov-Zakharov solutions (4–6) have clear physical meaning of total fluxes of wave action (5), energy (4) and momentum (6) and do not refer to specific initial conditions. This is not the case for the higher-order

terms. The link between these additional terms with inherent properties of the collision integral $S_{nl}$ and/or with specific initial conditions is a subject of further studies.

## 2.2 Self-similar solutions of the kinetic equation

The homogeneity property (2) is extremely useful for studies of non-stationary (inhomogeneous) solutions of the kinetic equation. Approximate self-similar solutions for reference cases of duration- and fetch-limited development of wave field can

be obtained under the assumption of dominance of the wave-wave interaction term $S_{nl}$ (Pushkarev et al., 2003; Zakharov, 2005; Badulin et al., 2005a; Zakharov and Badulin, 2011). These solutions exhibit the so-called incomplete or the second type self-similarity (e.g. Barrenblatt, 1979). In terms of frequency-angle dependencies of wave action spectra one has for the duration- and fetch-limited cases correspondingly (Badulin et al., 2005a, 2007; Zakharov et al., 2015)

$$N(\omega, \theta, \tau) = a_\tau \tau^{p_\tau} \Phi_{p_\tau}(\xi, \theta) \tag{8}$$

$$N(\omega, \theta, \chi) = a_\chi \chi^{p_\chi} \Phi_{p_\chi}(\zeta, \theta) \tag{9}$$

with dimensionless time $\tau$ and fetch $\chi$

$$\tau = t/t_0; \qquad \chi = x/x_0. \tag{10}$$

Dimensionless arguments of shape functions $\Phi_{p_\tau}(\xi)$, $\Phi_{p_\chi}(\zeta)$ in (8,9) contain free scaling parameters $b_\tau$, $b_\chi$ and exponents of frequency downshifting $q_\tau$, $q_\chi$

$$\xi = b_\tau \omega^2 \tau^{-2q_\tau}; \qquad \zeta = b_\chi \omega^2 \chi^{-2q_\chi}. \tag{11}$$

Homogeneity property (2) dictates 'magic relations' (in the words of Pushkarev and Zakharov, 2015, 2016) between exponents $p_\tau$, $q_\tau$ and $p_\chi$, $q_\chi$

$$p_\tau = \frac{9q_\tau - 1}{2}; \qquad p_\chi = \frac{10q_\chi - 1}{2}. \tag{12}$$

Additional 'magic relations' coming from homogeneity property (2) fix a link between the amplitude scales $a_\tau$, $a_\chi$ and the
bandwidth scales $b_\tau$, $b_\chi$ of the self-similar solutions (8–11)

$$a_\tau = b_\tau^{19/4}; \qquad a_\chi = b_\chi^{5/2}. \tag{13}$$

Thus, 'magic relations' (12,13) reduce number of free parameters of the self-similar solutions (8,9) from four (two exponents and two coefficients) to two only: a dimensionless exponent $p_\tau$ ($p_\chi$) and an amplitude of the solution $a_\tau$ ($a_\chi$).

The shape functions $\Phi_{p_\tau}(\xi, \theta)$, $\Phi_{p_\chi}(\zeta, \theta)$ in (8,9) are specified by solutions of a nonlinear boundary problem for an integro-
differential equation in self-similar variables $\xi$ or $\zeta$ (conditions of decay at zero and infinity) and angle $\theta$ (periodicity) (see sect. 5.2 Badulin et al., 2005a, for details). These solutions reveal relatively narrow angular distributions with a single pronounced maximum and remarkably weak dependence on exponent of wave growth $p_\xi$, ($p_\chi$) as simulations show (e.g. Badulin et al., 2008a). This feature of *quasi-universality* (in the words of Badulin et al., 2005a) of the solutions of nonlin-ear problem can be treated within a diffusion approximation for the kinetic equation (Zakharov and Pushkarev, 1999, see
also Zakharov (2010)) as a 'survival' of very few eigen-functions – angular harmonics of the corresponding linear boundary problem. As it will be shown below the weakly anisotropic KZ solution (6) represents a principal angular harmonic of such decomposition.

Two-lobe patterns can be observed beyond the spectral peak as local maxima at oblique directions or as 'shoulders' in wave frequency spectra. Their appearance within the kinetic equation approach is generally associated with wave genera-
tion by wind (e.g. Bottema and van Vledder, 2008, 2009) and/or effect of wave-wave interactions (Banner and Young, 1994; Pushkarev et al., 2003). Numerical simulations within the potential Euler equations also show formation of the two-lobe pat-terns for rather short times (a few hundreds of spectral peak periods) of evolution of initially unimodal spectral distribution (Toffoli et al., 2010).

An essential approximation which is widely used both for experimentally observed and simulated wave spectra is generally
treated as an important property of *spectral shape invariance* (terminology of Hasselmann et al., 1976) or *the spectra quasi-universality* (in the words of Badulin et al., 2005a). In fact, such 'invariance' does not suppose a point-by-point matching of properly normalized spectral shapes. Proximity of integrals of the shape functions $\Phi_{p_\tau}$, $\Phi_{p_\chi}$ in a range of wave growth rates $p_\tau$, $p_\chi$, appears to be sufficient, in particular, for formulating efficient semi-empirical parameterizations of wind-wave growth in terms of integral values (e.g. Hasselmann et al., 1976). Consistent analysis within the weak turbulence approach that used
this important approximation has recently lead to a remarkable theoretically-based relationship (Zakharov et al., 2015)

$$\mu^4 \nu = \alpha_0^3. \tag{14}$$

Here wave steepness $\mu$ is estimated from total wave energy $E$ and spectral peak frequency $\omega_p$

$$\mu = \frac{E^{1/2}\omega_p^2}{g}. \tag{15}$$

The 'number of waves' $\nu$ in a spatially homogeneous wind sea (i.e. for duration-limited case) is defined as follows:

$$\nu = \omega_p t. \tag{16}$$

For spatial (fetch-limited) wave growth, the coefficient of proportionality $C_f$ in the equivalent expression $\nu = C_f |\mathbf{k}_p| x$ ($\mathbf{k}_p$ being the wavevector of the spectral peak) is close to the ratio between the phase and group velocities $C_{ph}/C_g = 2$. A universal constant $\alpha_0 \approx 0.7$ is a counterpart of the constants $C_p, C_q$ of the stationary Kolmogorov-Zakharov solutions (4,5) and has a similar physical meaning of a ratio between wave energy and the energy spectral flux (in power $1/3$). A remarkable feature of the universal wave growth law (14) is its independence of wind speed. This wind-free paradigm based on intrinsic scaling of wave development is shown to be a useful tool of analysis of wind-wave growth (Zakharov et al., 2015). Below we demonstrate its effectiveness for interpreting swell simulations.

## 2.3   Self-similarity of swell solutions

The self-similar solution for swell is just a member of a family of solutions (8,9) with special values of temporal or spatial rates

$$p_\tau = 1/11; \qquad q_\tau = 1/11 \tag{17}$$
$$p_\chi = 1/12; \qquad q_\chi = 1/12 \tag{18}$$

Exponents (17,18) provide conservation of the total wave action for its evolution in time (duration-limited setup) or in space (fetch-limited)

$$N = \int_0^\infty N(\omega,\theta)d\omega d\theta = \text{const} \tag{19}$$

On the contrary, total energy

$$E = \int \omega N(\mathbf{k})d\mathbf{k} \tag{20}$$

and wave momentum

$$\mathbf{K} = \int \mathbf{k} N(\mathbf{k})d\mathbf{k} \tag{21}$$

are only formal constants of motion of the Hasselmann equation and decay with time $t$ or fetch $x$

$$E \sim t^{-1/11}; \qquad K_x \sim t^{-2/11} \tag{22}$$
$$E \sim x^{-1/12}; \qquad K_x \sim x^{-2/12}. \tag{23}$$

The swell decay (22,23) reflects a basic feature of the kinetic equation for water waves: energy (20) and momentum (21) are not conserved (see Zakharov et al., 1992; Pushkarev et al., 2003, and refs. herein). The wave action is the only true integral of the kinetic equation (1).

The swell solution manifests another general feature of evolving spectra: the downshifting of the spectral peak frequency (or other characteristic frequency), i.e.

$$\omega_p \sim t^{-1/11}; \qquad \omega_p \sim x^{-1/12}. \tag{24}$$

The universal law of wave evolution (14) is, evidently, valid for the self-similar swell solution as well with a minor difference in the value of the constant $\alpha_0$. As soon as this constant is expressed in terms of the integrals of the shape functions $\Phi_\tau$, $\Phi_\chi$ and the swell spectrum shape differs essentially from ones of the growing wind seas, this constant appears to be less than $\alpha_0$ of the growing wind seas.

The theoretical background presented above is used below for analysis of results of simulations.

## 3   Swell simulations

### 3.1   Simulation setup

Simulations of ocean swell require special care. First of all, calculations for quite long periods of time (up to $2 \cdot 10^6$ seconds in our case) should be accurate enough in order to capture relatively slow evolution of solutions and, thus, be able to relate results with the theoretical background presented above. Duration-limited evolution of the swell has been simulated with the Pushkarev et al. (2003) version of the code based on the WRT algorithm (Webb, 1978; Tracy and Resio, 1982). Features of the code and numerical setups have been described in previous papers (Badulin et al., 2002, 2004, 2005a, b, 2007; Zakharov et al., 2007; Badulin et al., 2008a, 2013; Pushkarev and Zakharov, 2015, 2016). Frequency resolution for log-spaced grid has been set to $(\omega_{n+1} - \omega_n)/\omega_n = 1.03128266$. It corresponds to 128 grid point in frequency range $0.02 - 1$ Hz (approximately $1.5$ to $3850$ meters wave length).

Standard angular resolution $\Delta\theta = 10°$ has been taken as adequate for the goals of our study. A control series of runs with angular resolution $\Delta\theta = 5°$ showed very close but still quantitatively different shaping of wave spectra (see discussion below) while differences of integral parameters (wave height, period, total momentum) did not exceed $1\%$ after $2 \cdot 10^6$s of evolution.

Initial conditions were similar in all series of simulations: spectral density of action in wavenumber space was almost constant in a box of the wavenumber modulo and angles. Slight modulation ($5\%$ of the box height) and low pedestal outside the box (six orders less than the maximal value) have been set in order to stimulate wave-wave interactions since the collision integral $S_{nl}$ vanishes for $N(\mathbf{k}) = \text{const}$:

$$N(\mathbf{k}) = \begin{cases} N_0(1 + 0.05\cos^2(\theta/2)), \ |\theta| < \Theta/2, \omega_l < \omega < \omega_h \\ 10^{-6}N_0, \qquad \text{otherwise} \end{cases} \tag{25}$$

In (25) the references to angle $\theta$ ($\cos\theta = k_x/|\mathbf{k}|$) and wave frequency $\omega$ are used for conciseness of the expression for spatial wave action spectrum $N(\mathbf{k})$. The default values $\omega_l$ and $\omega_h$ corresponding to wave periods 10 and 2.5s have been used for the most cases providing sufficient space for spectral evolution to low frequencies (spectra downshifting) and for stability of calculations at high frequencies for the default cutoff frequency $f_c = 1$Hz.

Dissipation was absent in the runs. Free boundary conditions were applied at the high-frequency end of the domain of calculations: generally, short-term oscillations of the spectrum tail do not lead to instability, i.e. the resulting solutions can be regarded as ones corresponding to conditions of decay at infinitely small scales ($N(\mathbf{k}) \to 0$ when $|\mathbf{k}| \to \infty$).

Calculations with a hyper-viscosity (e.g. Pushkarev et al., 2003) or a diagnostic tail at the high-frequency range of the spectrum (Gagnaire-Renou et al., 2010) do not affect results quantitatively compared to our simulations without any dissipa-
tion. Thus, these 'non-conservative' options can mimic successfully the effect of energy leakage at $|\mathbf{k}| \to \infty$ in our formally non-dissipative problem. Very strong dissipation at less than 10 grid points at the very end of frequency domain suppresses spectral level and, simultaneously, reduces the overall energy dissipation at these points. Thus, the effect on the evolution of the energy-containing part of the solution appears to be quite weak and depends slightly on particular form and magnitude of the hyper-viscosity. In some cases, the hyper-viscosity option that suppresses high-frequency noise can accelerate calculations.
In a sense, it is equivalent to reducing an effective number of grid points. Test runs with the reduced frequency domain (cutoff up to $f_c = 0.6$Hz, 112 grid points) did not show essential quantitative difference with the default option ($f_c = 1$Hz, 128 grid points).

In contrast to wind-driven waves where wind speed is an essential physical parameter that gives a useful physical scale, the swell evolution is determined by initial conditions only, i.e. by $N_0$ (dimension of wave action spectral density $[N(\mathbf{k})] =$
[Length$^4$·Time]), a characteristic frequency (sideband $[\omega_l, \omega_h]$) and angular spreading $\Theta$ within the setup (25). We tried different combinations of these parameters. Three frequency bands $[0.026 - 0.09]$, $[0.058 - 0.25]$, $[0.1 - 0.4]$ Hz have been chosen to generate swell with wavelengths approximately $200, 300, 400$ meters at final stages of evolution. The angular spreading $\Theta$ was set at $30°, 50°, 170°, 230°$ and $330°$. Initial significant wave heights $H_s$ were taken as approximately $4.8, 8, 10, 12, 18$ meters. As it will be detailed below an abrupt fall of wave energy occurred at the very first hours of evolution (up to $50\%$ for the first 1
hour). Thus, the above high values of $H_s$ can be accepted as realistic values for sea swell. Totally, more than 30 combinations of wave height, frequency range and angular spreading have been simulated for the duration at least $10^6$ s. In some cases, for high amplitudes and narrow angular spreadings, simulations have failed because of strong numerical instability.

Below we focus on the series of Table 1 where initial wave heights were fixed (within $2\%$) at approximately 4.8 meters and angular spreading varied from very narrow $\Theta = 30°$ to almost isotropic $\Theta = 330°$ (25). The frequency range of the initial
perturbations was $0.1 - 0.4$Hz. The simulations have been carried out for duration $2 \cdot 10^6$ seconds with angular resolution $\Delta\theta = 10°$ and checked for series sw030 and sw330 with $\Delta\theta = 5°$.

## 3.2 Self-similar features of swell

Evolution of swell spectra with time is shown in fig.1 for the case sw330 of Table 1. The example shows a strong tendency to self-similar shaping of wave spectra. This remarkable feature has been demonstrated and discussed for swell in previous works

(Badulin et al., 2005a; Benoit and Gagnaire-Renou, 2007; Gagnaire-Renou et al., 2010) for special parameters that provided relatively fast evolution of rather short and unrealistically high waves. In our simulations, we start with the mean wave period of about 3 seconds that corresponds to the end of calculations of Badulin et al. (2005a, see fig. 8 therein). The initial spectrum evolves very quickly and keeps a characteristic shape for less than 1 hour when wave steepness falls dramatically below

$\mu = 0.15$ ($T_p \approx 6$s) while wave height looses only about 20% of its initial value (see fig.1, green curve for $t = 0.6$ hours). For 555 hours the spectral peak period reaches 11.4 seconds (the corresponding wavelength $\lambda \approx 200$ meters) and wave steepness becomes $\mu = 0.022$. The final significant wave height $H_s \approx 2.8$ meters is essentially less than its initial value 4.8 meters. All these values can be considered as typical ones for ocean swell.

Dependence of key wave parameters on time is shown in fig. 2 for different runs of Table 1. Power-law dependencies of self-

similar solutions (17,18,22-24) are shown by dashed lines. In fig. 2$a,b$ total wave energy $E$ and the spectral peak frequency $\omega_p$ show good correspondence to power laws of the self-similar solutions (8). By contrast, power-law decay of $x-$component of wave momentum $K_x$ depends essentially on angular spreading of initial wave spectra. While for narrow spreading (runs sw030 and sw050) there is no visible deviation from the $K_x \sim t^{-2/11}$ law, wide-angle cases clearly show these deviations. The 'almost isotropic' solution for sw330 is tending quite slowly to the theoretical dependency of wave momentum $K_x$ (23).

The duration more than 3 weeks appears 'too short': one can see a transitional behavior when wave spectra evolve from the 'almost isotropic' state to an inherent distribution with a pronounced anisotropy.

A simple quantitative estimate of the 'degree of anisotropy' is given in fig.2$d$. Evolution of dimensionless parameter of anisotropy in terms of the approximate Kolmogorov-Zakharov solution (6) by Katz and Kontorovich (1974) is shown for all the cases of Table 1. We introduce parameter of anisotropy $A$ as follows

$$A = \frac{gM}{\omega_p P}. \tag{26}$$

where total energy flux $P$ (energy flux at $\omega \to \infty$) is estimated from evolution of total energy

$$P = -\frac{dE}{dt}. \tag{27}$$

Similarly, total wave momentum (21) provides an estimate of its flux as follows

$$M = -\frac{dK_x}{dt}. \tag{28}$$

Spectral peak frequency $\omega_p$ has been used for the definition of 'degree of anisotropy', $A$ (26). Different scenarios are seen in fig. 2$d$ depending on angular spreading of wave spectra. Nevertheless, a general tendency to a universal behavior at very large times (more than $2 \cdot 10^6$ seconds) looks quite plausible.

Similar dispersion of runs depending on anisotropy of initial distributions is seen in fig. 3 when tracing the invariant of the self-similar solutions (14). Again, like in fig.2b, $2 \cdot 10^6$ seconds are not sufficient to demonstrate validity of relationship (14)

in its full. A limit $\alpha_0$ (14) is very likely reached at larger times. This limit is a bit less (by approximately $15\%$) than one for growing wind seas $\alpha_0 \approx 0.7$. Again, the 'almost isotropic' solution shows its stronger departure from the rest of the series. The differences are better seen in angular distributions rather than in normalized spectral shapes (fig. 4) when we are trying to check self-similarity features of the solutions in the spirit of Badulin et al. (2005a); Benoit and Gagnaire-Renou (2007).

### 3.3 Directional spreading of swell spectra

Despite significant difference of the runs in integral characteristics of the swell anisotropy (e.g. figs. 2*b,d*), the resulting spectral distributions still show pronounced features of universality as it is seen in frequency spectra (fig.4). As it will be shown below this universality of swell spectra is seen in angular distributions as well. This is of importance in the context of remarks of

sect.2.2: while the shape functions $\Phi_{p_\tau}$, $\Phi_{p_\chi}$ of self-similar solutions (8,9) are not unique there is likely a mechanism of their selection that supports the universality of the swell spectral distributions. Within a linear theory, it could be treated as survival of the only eigenfunction or, more prudently, of very few eigenmodes of the problem. As mentioned in sect.2.2. this 'linear' treatment can be used with some reservations for our problem which is heavily nonlinear in terms of wave spectra but allows for a quasi-linear analysis in terms of spectral fluxes (see Zakharov and Pushkarev, 1999; Pushkarev et al., 2003).

The only physical mechanism of the mode selection in the swell problem is nonlinear relaxation to an inherent state due to four-wave resonant interactions. This relaxation generally occurs at essentially shorter time scales than ones of wind pumping and wave dissipation (Zakharov and Badulin, 2011). There is no contradiction with the today vision of the sea wave balance in the above statement. The effect of nonlinear interactions on wave spectra is two-fold: firstly, it supports an inherent shaping of the spectra by very fast feedback to its perturbation and, secondly, it is responsible for relatively slow nonlinear cascading

within this inherent shaping.

Normalized sections of spectra at the peak frequency $\omega_p$ are shown in fig.5 for runs of Table 1 at $t = 10^6$ seconds (approx. 11.5 days). 'The almost isotropic' run sw330 shows relatively high pedestal of about 2% of maximal value while other series have a background one more order less. At the same time, the core of all distributions is quite close to a gaussian shape

$$y_{gauss} = \exp\left(-\frac{\theta^2}{2\sigma^2}\right) \tag{29}$$

with half-width $\sigma = 35°$ (dashed curve in fig.5). Experimentally based spreading functions are represented in fig.5 by two reference curves. For growing wind seas the dependence by Donelan et al. (1985, eq.9.2)

$$y_{1985} = \text{sech}^2(\beta\theta); \quad \beta = 2.28 \tag{30}$$

gives almost twice narrower distribution (dot line in fig.5).The wrapped-normal fit of angular distribution for one of the case of the West Africa Swell Project (see Table 11.2 and fig.11.8 in Ewans et al., 2004) with standard deviation $\sigma \approx 14.3°$ gives a

sharper distribution shown by a dashed curve.

Evolution of directional spreading in time is shown in absolute values in fig. 6 for three runs: the most anisotropic case sw030 (fig. 6*a,b*), weakly anisotropic initial state sw230 (fig. 6*c,d*) and 'the almost isotropic' run sw330 (fig. 6*e,f*). In the left column the angular spreading at peak frequency shows remarkably close patterns for the first two cases: peak values at large times differ by few percents only. The weakly anisotropic case sw230 (initial angular spreading 230° with essential

counter-propagating fraction) reaches its almost saturated state for a couple of days only (cf. curves at $t = 17$ and $t = 35$ hours). Similar proximity of these two cases can be observed for integrals of spectra in frequency as shown in the right column of fig.6,

i.e. for values

$$\mathcal{E}(\theta) = \int_0^{\omega_c} E(\omega,\theta)d\omega. \tag{31}$$

Self-similar solutions (8) predict a power-law decay of magnitude of $\mathcal{E}$ with time which is what we see in fig.6*b,d* for the first two cases. Behavior of 'the almost isotropic' case `sw330` is qualitatively different. The relatively strong adjustment to a narrow directional spreading occurs in course of all the duration $2 \cdot 10^6$ s. The duration appears to be too short to reach a self-similar regime resembling cases `sw030`, `sw230`.

The effect of sharpening of angular distributions of the run `sw330` in fig.6*e,f* requires additional comments. First, it manifests a transitional nature of the case `sw330` when a solution is rather far from its self-similar asymptotics. Secondly, this case illustrates the above statement of the paragraph on two scales of wave spectra evolution. The angular adjustment occurs at relatively short temporal scales as compared with slow evolution of integral parameters (cf. fig.2). This adjustment is provoked by excursion of initially 'almost isotropic' distribution from an anticipated 'inherent state' that, thus, stimulates wave-wave interactions as a mechanism of relaxation. The example demonstrates ability of wave-wave interactions to effectively rebuild directional distributions. Note, that in some cases, say, in the problem of relaxation of wave field to sudden changes of wind direction the wave-wave interactions are considered as ineffective as compared to relaxation 'due mainly to imbalance $S_{in} < S_{diss}$' (e.g. Young et al., 1987, $S_{in}$ – wind input, $S_{diss}$ – wave dissipation).

### 3.4 Bi-modality of swell spectra

Bi-modality of directional spreading of ocean swell is widely discussed for experimental data as a possible result of swell evolution (e.g. Ewans, 1998, 2001; Ewans et al., 2004). Our simulations encounter this effect as a persistent feature of swell spectra. Fig.7 represents directional spreading of swell spectra in two ways. The left column shows directional distribution function $H(\omega,\theta)$ in the spirit of widely used definition (e.g. Ewans, 1998)

$$E(\omega,\theta) = \bar{E}(\omega)H(\omega,\theta), \quad H(\omega,\theta) \geq 0, \quad \int_{-\pi}^{\pi} H(\omega,\theta)d\omega = 1. \tag{32}$$

An alternative representation in the right column of fig.7 uses spectral densities normalized by their maxima at fixed frequency to trace 'ridges' of surface $\tilde{E}(\omega,\theta)$ defined as follows (cf. eq.1 in Young et al., 1995)

$$\tilde{E}(\omega,\theta) = E(\omega,\theta)/\max_{-\pi<\theta\leq\pi}\left(E(\omega,\theta)\right). \tag{33}$$

Both representations reveal bi-modality of swell spectra fairly well for all cases of Table 1. 'Narrow' initial spectrum `sw030` and 'wide' one `sw170` evolves to very close X-shaped side-lobe patterns (fig.7*a,c*). Pronounced side-lobes are seen both above and below the spectral peak frequency. Directional distribution function $H(\omega,\theta)$ (32) does not show similar pattern for 'the almost isotropic' case `sw330` (fig.7*e,g*) but the X-shapes are seen fairly well in the 'ridge' representation (33) for all the cases. Directional spreading for the run `sw330` is shown for simulations with standard angular resolution $\Delta\theta = 10°$ (fig.7*e,f*)

and with fine one $\Delta\theta = 5°$ (fig.7$g,h$). Higher resolution makes 'ridges' sharper and allows for resolving more details of the directional distribution. In particular, side-lobes appear for counter-propagating waves at $\theta \approx \pm 3\pi/4$ and $\omega/\omega_p \approx 5/4$. At the same time, the standard angular resolution in our simulations $\Delta\theta = 10°$ seems to be adequate for the bi-modality phenomenon.

The patterns similar to ones of fig.7 have been obtained in simulations of the Hasselmann equation for wind-driven waves with the exact term of nonlinear transfer $S_{nl}$ by Banner and Young (1994); Young et al. (1995) at formally finer resolution $\Delta\theta = 6.67°$. It should be noted that directions beyond the cone $\theta = \pm 120°$ have not been taken into account to speed up calculations in the cited papers. It can explain discrepancy with our results at the high frequency end of fig.7$f,h$ (cf. Plate 1 in Young et al., 1995). This point can be clarified in further studies.

An important issue of agreement of our results and findings of Banner and Young (1994); Young et al. (1995) is presence of
low-frequency (below the spectral peak) side-lobes. Experimental results by Ewans (cf. figs.8,16 1998) show good correspondence of the directional spreading functions with numerical results at high frequencies but do not fix any side-lobes below the spectral peak.

Generally, the phenomenon of side-lobe occurrence is associated with a joint effect of wave-wave interactions and wave generation by wind (e.g. Banner and Young, 1994; Pushkarev et al., 2003; Bottema and van Vledder, 2008). The theoretical
background of sect. 2.1 and our simulations of swell can propose an interpretation and alternative ways of advanced analysis of the effect in terms of stationary solutions of Kolmogorov-Zakharov (7). These solutions being presented as power series of dimensionless ratios of spectral fluxes and as an extension of the approximate solution (6) by Katz and Kontorovich (1974) predict higher-order angular harmonics and can be found within the formal procedure of Pushkarev et al. (2003, 2004). This approach is not fully correct in the vicinity of the spectral peak but still looks plausible and useful for interpretation of the
effect of wave-wave interactions. Analysis of the next paragraph shows perspectives of the KZ solution paradigm.

### 3.5 Swell spectra vs KZ solutions

Very slow evolution of swell in our simulations provides a chance to check relevance of the classic Kolmogorov-Zakharov solutions (4-7) to the problem under study. The key feature of the swell solution from the theoretical viewpoint is its 'hybrid' (in the words of Badulin et al., 2005a) nature: the inverse cascade (negative fluxes) determines evolution of spectral peak and
its downshifting, while the direct cascade (positive fluxes) occurs at frequencies slightly (approximately 20%) above the peak. This hybrid nature is illustrated by fig. 8 for energy and wave momentum fluxes. In order to avoid ambiguity in treatment of the simulation results within the weak turbulence theory we will not discuss this hybrid nature of swell solutions and focus on the direct cascade regime. Thus, general solution (7) in the form

$$E(\omega,\theta) == \frac{P^{1/3}g^{4/3}}{\omega^4}G(0, gM/(\omega P), \theta)$$

and its approximate explicit version (6) by Katz and Kontorovich (1971, 1974) will be used below for describing the direct cascading of energy and momentum at high frequency (as compared to $\omega_p$) .

Two runs of Table 1, `sw030` and 'almost isotropic' `sw330`, are presented in fig.8 in order to show qualitative similarity of extreme cases of initial directional spreading. Positive fluxes $P$ and $M$ decays with time in good agreement with power-law

dependencies (22) and have rather low variations in relatively wide frequency range $3\omega_p < \omega < 6\omega_p$ in fig. 8. For energy fluxes $P$ (figs.8$a,b$) one can see good quantitative correspondence (note, that times for some curves are slightly different). Absolute values of momentum flux $M$ as well as magnitudes of wave momentum itself (see fig.2) differ by more than one order.

The domain of quasi-constant fluxes $\omega > 3\omega_p$ can be used for verification of relevance of the stationary KZ solutions (4–6) to the quasi-stationary swell solutions. All the cases of Table 1 show very close patterns of spectral fluxes (e.g. fig.8) and, what is more important, very close estimates of Kolmogorov's constants.

The first and the second Kolmogorov's constants can be easily estimated for the approximate solution (6) from combinations of along- and counter-propagating spectral densities as follows

$$C_p = \frac{\omega^4 \left( E(\omega,0) + E(\omega,\pi) \right)}{4g^{4/3}P^{1/3}} \tag{34}$$

$$C_m = \frac{\omega^5 P^{2/3} \left( E(\omega,0) - E(\omega,\pi) \right)}{4g^{7/3}M}. \tag{35}$$

These estimates provide very close values of the Kolmogorov constants for all the series of Table 1 with the only exception of 'the almost isotropic' run sw330 for the second Kolmogorov constant $C_m$. Fig. 9 gives the first Kolmogorov constant $C_p \approx 0.21 \pm 0.01$ (slightly lower values for initially narrow distributions) and $C_m \approx 0.08 \pm 0.02$ for all the runs except sw330 (cf. figs.9$b,d$ for 'narrow' sw030 and 'wide' sw230).

The analytic estimate gives very close result $C_p = 0.219$ (Zakharov, 2010, eq.4.33). Numerical simulations by Lavrenov et al. (2002); Pushkarev et al. (2003); Badulin et al. (2005a) missed a factor of 2 in definitions of the Kolmogorov constants (cf. our definitions 4-6 and eqs. 4.29, 4.30 in Zakharov, 2010). Taking this into account, one has the reported values $0.151 < C_p < 0.162$; $0.105 < C_m < 0.121$ in Lavrenov et al. (2002, Table 1), $0.16 < C_p < 0.23$; $0.09 < C_m < 0.14$ in Pushkarev et al. (2003, eqs. 5.3, 5.6, 5.8) and $0.19 < C_p < 0.20$ in Badulin et al. (2005a). The first experimental attempt to evaluate the first Kolmogorov constant by Deike et al. (2014) presented value $C = 1.8 \pm 0.2 \approx 2\pi C_p$, i.e. $2\pi$ times bigger counterpart of $C_p$.

While the estimates of the Kolmogorov's constants for the swell look consistent the numerical solutions differ essentially from the approximate weakly anisotropic KZ solution (6). The directional spreading cannot be described by the only angular harmonics as in (6), higher-order corrections are clearly seen in figs.7 as side-lobes. Nevertheless, the robustness of the estimates of the second Kolmogorov constant $C_m$ provides a good reference for estimates of the spectra anisotropy.

The estimates of $C_m$ for sw330 (fig. 9$f$) demonstrate a specific nonstationarity of the swell solution in terms of wave momentum flux while the first Kolmogorov constant $C_p$ (fig.9$f$) show relevance of the stationary KZ solutions to the swell problem.

## 4  Discussion. Swell and ocean environment

Results of our simulations showed their fairly good correspondence to findings of the theory of wave (weak) turbulence. Relevance of these results to experimental facts seems to be a logical close of this work. The issue of relevance is two-fold. First, our results can help in explaining effects which interpretation in terms of alternative approaches (mostly, within linear

theory) is questionable. Secondly, one can formulate, or, at least, sketch cases where our approach becomes invalid or requires an extension. Both aspects are considered in the final section.

Attenuation in course of long term swell evolution is an appealing problem of the swell monitoring. We show that contribution of wave-wave interactions to this process can be important mostly at initial stages of swell evolution. The observed rates

of swell attenuation in an open ocean cannot be treated within our approach for a number of reasons. First of all, the duration-limited setup of our simulations do not account for important mechanisms of frequency dispersion and spatial divergence due to sphericity of the Earth. These mechanisms can both contribute into swell attenuation together with wave-wave interactions and essentially contaminate results of observations. The intrinsic swell attenuation is, generally, small as compared to the effect of reduction (or amplification at large fetches) (see fig.2b in Ardhuin et al., 2009) which is accounted for within the linear model

of geometrical optics whose validity is generally assumed for ocean swell.

Ocean swell for long times (fetches), becomes likely an important constituent of the ocean environment which can be heavily affected by relatively short wind-driven waves. We discuss the effect of swell amplification at rather low wind speeds and give tentative estimates based on the approach of this paper.

### 4.1   Swell attenuation within the kinetic equation

Dependence of wave height on time is shown in upper panel of fig. 10 (see also fig.2) for the runs of Table1. All the runs show quantitatively close evolution. Strong drop of up to $30\%$ of initial value occurs within a relatively short time of about one day. An essential part of the wave energy leakage corresponds to this transitional stage at the very beginning of swell evolution when swell is tending very rapidly to self-similar asymptotics. Afterwards, the decay becomes much slower following the power-law dependence of the self-similar solutions (22).

For comparison with other models, and available observations, the duration-limited simulations have been recasted into dependencies of fetch through the simplest time-to-fetch transformation (e.g. Hwang and Wang, 2004; Hwang, 2006):

$$x(s) = \int\limits_0^s C_g(\omega_p(t))dt. \tag{36}$$

The equivalent fetch is estimated as a distance covered by a wave guide travelling with the group velocity of the spectral peak component. The corresponding dependencies are shown in bottom panel in fig.10. Two quasi-linear models by Ardhuin et al.

(2009) and Babanin (2006) predict relatively slow attenuation at fetches in a 'near zone' less than 1000 km (approximately 1 day) and then gradual decay up to very few of the percentage points of initial value at final distances about 18000 km where our model shows qualitatively different weak attenuation.

It should be noted that our model describes attenuation of the ocean swell 'on its own' due to wave-wave interactions without any external effects. Thus, the effect of an abrupt drop of wave amplitude at short time (fetch) should be taken into consideration

above all others when discussing possible application of our results to swell observations and physical interpretation of the experimental results.

## 4.2 Swell and wind sea coupling. Arrest of weakly turbulent cascading

Extremely weak attenuation of swell due to wave-wave interactions provokes a question on robustness of this effect. A variety of physical mechanisms in the ocean environment can change the swell evolution qualitatively. The above discussion of swell attenuation presents a remarkable example of such transformation when dissipation becomes dominant. Tracking of swell events from space gives an alternative scenario of transformation when swell appears to be growing. Satellite tracks can comprise up to $30\%$ of cases of growing swell 'most of them are not statistically significant' (Jiang et al., 2016). Nevertheless, a possible effect of wind-sea background on long ocean swell opens an important discussion in view of theoretical (Badulin et al., 2008b) and experimental (Benilov et al., 1974; Badulin and Grigorieva, 2012) results that demonstrate swell amplification by wind wave background.

As noted and shown above, evolution of swell can occur at different time scales for different physical quantities. Integrals of motion (energy, action, momentum) evolve at relatively large scales: frequency downshift and energy follows power-law dependencies $1/11$ ($\omega_p \sim t^{-1/11}$ and $E \sim t^{-1/11}$). The slow evolution is supported by interactions within a wave spectra that is close to an 'inherent' quasi-stationary state.

Oppositely, spectral shaping is evolving due to excursions from an 'inherent state' at much shorter scales that can be estimated following Zakharov and Badulin (2011, see eqs.21,22 therein). The nonlinear relaxation rate as defined by eqs.14-16 of the cited paper can be written as

$$\Gamma(\omega) = B\omega \left(\frac{\omega}{\omega_p}\right)^3 \mu^4 H(\omega, \theta). \tag{37}$$

Here $B$ is a big dimensionless coefficient (e.g. $B = 22.5\pi \approx 70.7$ for an isotropic spectrum, see Zakharov and Badulin, 2011) and $H(\omega, \theta)$ is directional distribution function (32). The big coefficient $B$ in (37) provides relatively fast relaxation of local excursions (in wave scales) from the slowly evolving 'inherent' swell, especially, in high frequency domain (factor $(\omega/\omega_p)^3$ in eq.37). Evidences of this relaxation can be seen in evolution of angular distribution of the run `sw330` where visible transformation of angular distribution is observed for all the duration of more than three weeks (fig.6): the non-self-similar background of the swell spectra is feeding the core of the spectral distribution.

A similar effect can be realized in the mixed sea when background of relatively short wind-driven waves feeds the swell. Total energy flux of the swell is decaying as rapidly as $dE/dt \sim t^{-12/11}$ and at sufficiently large time the associated direct cascading can be arrested by inverse cascading of wind-driven waves which fast relaxation to an 'inherent' swell ensures the swell feeding. This mechanism has been analyzed numerically (Badulin et al., 2008b) and showed its remarkable efficiency.

Simple estimates of possibility of the effect can be made in terms of balancing of two fluxes: direct cascade of swell and inverse cascade of wind-driven fraction. The swell energy leakage can be estimated from the weakly turbulent law (Badulin et al., 2007, eq.1.9) as follows

$$\left(\frac{dE}{dt}\right)_{direct} = \frac{E^3 \omega_p^9}{\alpha_{swell}^3 g^4} = \frac{\mu_{swell}^6 C_{swell}^3}{\alpha_{swell}^3 g} \tag{38}$$

Here swell parameters are marked by proper subscripts: $C_{swell} = g/\omega_p$ – phase velocity of the spectral peal component, $\mu_{swell}$ – swell steepness by definition (15), and $\alpha_{swell}$ – self-similarity parameter ($\alpha_{ss}$ in Badulin et al., 2007). Similar conversion

of sea state parameters to spectral flux can be done for the wind sea fraction (see sect.5.1 in Badulin et al., 2007, or Table 1 in Gagnaire-Renou et al. (2011))

$$\left(\frac{dE}{dt}\right)_{inverse} \approx C_w \left(\frac{\rho_a}{\rho_w}\right)^3 \frac{U_{10}^3}{\alpha_{wind}^3 g} \tag{39}$$

where coefficient $C_w = O(1)$ is introduced as soon as the conversion is based on dimensonal analysis and generalization of experimental results (Toba, 1972). A counterpart of $\alpha_{swell}$, the self-similarity parameter $\alpha_{wind}$ is approximately two times less in magnitude (Badulin et al., 2007). Thus, condition of balance of fluxes assotiated with different fractions of the mixed sea (38,39) says

$$2C_w \frac{\rho_a}{\rho_w} \frac{U_{10}}{C_{swell}} \approx \mu_{swell}^2 \tag{40}$$

For relatively short swell with period $T_p = 10$s ($\lambda \approx 150$m) and wind speed $U_{10} = 7$m/s one gets a critical swell steepness $\mu_{swell} \approx 0.03$. In other words, the mean-over-ocean wind 7m/s can balance (arrest) direct cascading of rather steep swell and, hence, provoke a growth of the swell due to absorbing short wind-driven waves. Evidently, this simple balance model gives very tentative estimate of the effect. Nevertheless, visual observations (Badulin and Grigorieva, 2012) and satellite data (Jiang et al., 2016), in our opinion, provide telling arguments for this phenomenon. Thus, 'negative dissipation' of swell (in the words of Jiang et al., 2016) could find its explanation within the simple model.

The simple estimate (40) shows a limited value of our 'pure swell' model for ocean environment. Potentially, the effect of even light wind on long-term propagation of swell can change the result qualitatively. Our pilot numerical studies (see also Badulin et al., 2008b) show importance of the swell and wind-sea coupling. This effect will be detailed in our further studies.

## 5 Conclusions

We presented results of sea swell simulations within the framework of the kinetic equation for water waves (the Hasselmann equation) and treated these properties within the paradigm of the theory of weak turbulence. A series of numerical experiments (duration-limited setup, WRT algorithm) has been carried out in order to outline features of wave spectra in a range of scales usually associated with ocean swell, i.e. wavelengths larger than 100 meters and duration of propagation up to $2 \cdot 10^6$ seconds (more than 23 days). It should be stressed that the exact collision integral $S_{nl}$ (nonlinear transfer term) has been used in all the calculations. Alternative, mostly operational approaches, like DIA (Discrete Approximation Approach) can corrupt the results quantitatively and even qualitatively.

Key results of the study:

1. A strong tendency for self-similar asymptotics is demonstrated. These asymptotics are shown to be insensitive to initial conditions in terms of evolution of integral quantities (wave energy, momentum). Moreover, universal angular distributions of wave spectra at large times have been obtained for both narrow (initial angular spreading $30°$) and almost isotropic initial spectra. Bi-modality of the spectral distributions in our simulations is found to be in agreement with previous numerical and experimental results (Banner and Young, 1994; Ewans, 2001; Ewans et al., 2004). The universality

of the spectral shaping can be treated as an effect of mode selection when very few eigenmodes of the boundary problem determines the system evolution. The inherent features of wave-wave interactions are responsible for this universality making the effect of initial conditions insignificant. Generally, the self-similar swell is co-existing with a background which is far from self-similar state;

2. The classic Kolmogorov-Zakharov (KZ) isotropic and weakly anisotropic solutions for direct and inverse cascades are shown to be relevant to slowly evolving sea swell solutions. Estimates of the corresponding KZ constants are found to agree well with previous analytical, numerical and experimental results. Thus, features of KZ solutions can be used as a reference for advanced approaches in the swell studies;

3. We show that an inherent peculiarity of the Hasselmann equation, energy and momentum leakage, can also be considered as a mechanism of the sea swell attenuation. The today models of sea swell are unlikely to account for this effect. Possible problems of the models are sketched in sect.3.1 when different options of simulation of the 'conservative dissipation' are discussed. All these options require sufficiently large high-frequency range where the short-term oscillations in absence of dissipation or hyper-viscosity can mimic the energy leakage at $|\mathbf{k}| \to \infty$. It should be noted, that the energy decay rates of sea swell in the numerical experiments, generally, do not contradict the results of recent swell observations and modelling. These studies based on satellite data and wave model hindcasting are focused mostly on 'far field' behavior of swell, generally, 1000 or more kilometers away from a stormy area. Our simulations show that a dramatic transformation of the swell occurs at shorter distances, in 'near field'. The essential swell energy losses in the near field, mostly due to nonlinear transfer, is an intriguing challenge for sea wave forecasting since the very first discussions of the phenomenon within the concept of wave-wave interactions (e.g. sects.8$a,b$ in Snodgrass et al., 1966). Thus, fig.10 outlines different domains of our model relevance rather than the model relevance for the general problem of ocean swell attenuation;

4. Long term evolution of swell is associated with rather slow frequency downshift ($\omega_p \sim t^{-1/11}$) and energy attenuation ($E \sim t^{-1/11}$). Meanwhile, the decay of other wave field quantities is essentially faster: wave steepness is decaying as $\mu \sim t^{-5/22}$ and total spectral flux even faster $dE/dt \sim t^{-12/11}$. This point is of key importance in our analysis as far as we consider nonlinear cascades of wave energy as governing physical mechanism of swell evolution. As we showed in discussion, the weak direct cascade of swell can be arrested by relatively light wind and then swell can start to grow. In our opinion, this conclusion correlates with manifestations of swell amplification in satellite data (Jiang et al., 2016) and in visual observations (Badulin and Grigorieva, 2012). Thus, 'negative dissipation' of swell (in the words of Jiang et al., 2016) could find its explanation within the simple estimate (40) of sect.4.2;

5. The last conclusion uncovers deficiency of the duration-limited setup for the phenomenon of swell. An alternative setup of fetch-limited evolution ($\partial/\partial t \equiv 0, \nabla_{\mathbf{r}} \neq 0$) introduces dispersion of wave harmonics as a competing mechanism that can change the swell evolution dramatically. Recent advances in wave modelling (Pushkarev and Zakharov, 2016) make the problem of spatial-temporal swell evolution feasible and specify the perspectives of our first step study. The theoretical background for the classic fetch-limited setup when solutions depend on the only spatial coordinate (i.e.

$\partial/\partial x \neq 0$, $\partial/\partial y \equiv 0$) is sketched in sect. 2 of this paper. The one-dimensional model add an essential physical effect of wave dispersion. A passage to polar coordinates allows us to consider an effect of spatial divergence in formally one-dimensional problem where solutions depend on radial coordinate but are still anisotropic in wavevector space. Self-similar solutions for this problem in the spirit of sect. 2 can be easily found and related to numerical results. All the prospective simulations require developing effective numerical approaches. In particular, high angular resolution (not worse than $5°$) could be recommended for these studies. V. Geogjaev & V. Zakharov has developed such code recently (a talk at the meeting Waves in Shallow Water Environment, 2016, Venice). We plan to use it in the swell studies.

*Acknowledgements.* Authors are thankful for the support of the Russian Science Foundation grant No. 14-22-00174. Authors are indebted to Prof. Victor Shrira and Vladimir Geogjaev for discussions and valuable comments. Authors are also grateful Dr. Andrei Pushkarev for his assistance in simulations. Authors appreciate critical consideration of the paper by reviewers: Dr. Gerbrant van Vledder and Dr. Sergei Annenkov. Their constructive feedbacks lead to substantial revision of sects.3 and 4.

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

**Table 1.** Initial parameters of simulation series

| ID | $\Theta$ | $N$ (m$^2 \cdot s$) | $H_s$ (m) |
|---|---|---|---|
| sw030 | 30° | 0.720 | 4.63 |
| sw050 | 60° | 0.719 | 4.6 |
| sw170 | 180° | 0.714 | 4.74 |
| sw230 | 240° | 0.721 | 4.67 |
| sw330 | 330° | 0.722 | 4.79 |

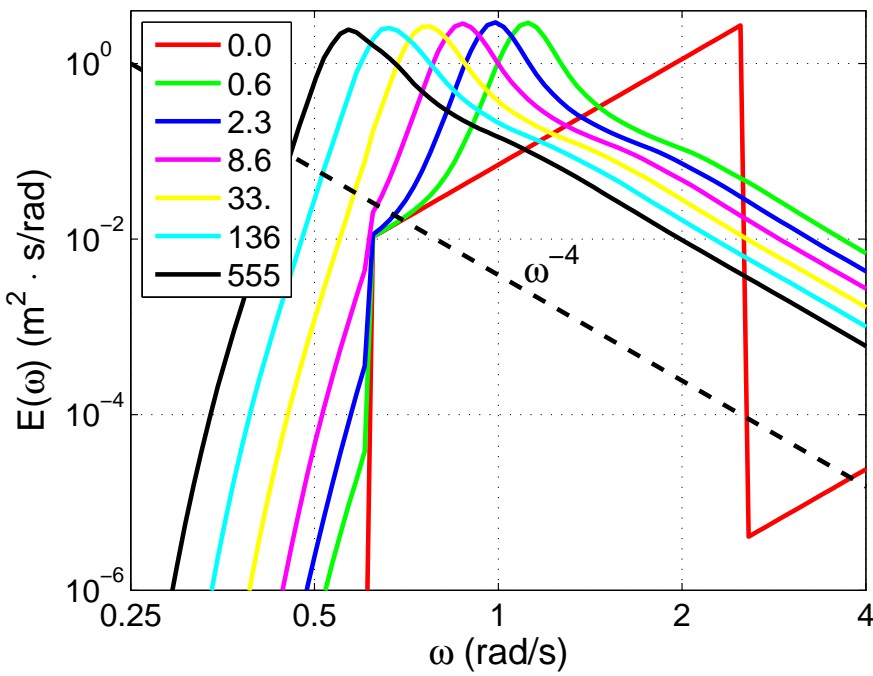

**Figure 1.** Frequency spectra of energy at different times (legend, in hours) for the case sw330 ($\Theta = 330°$).

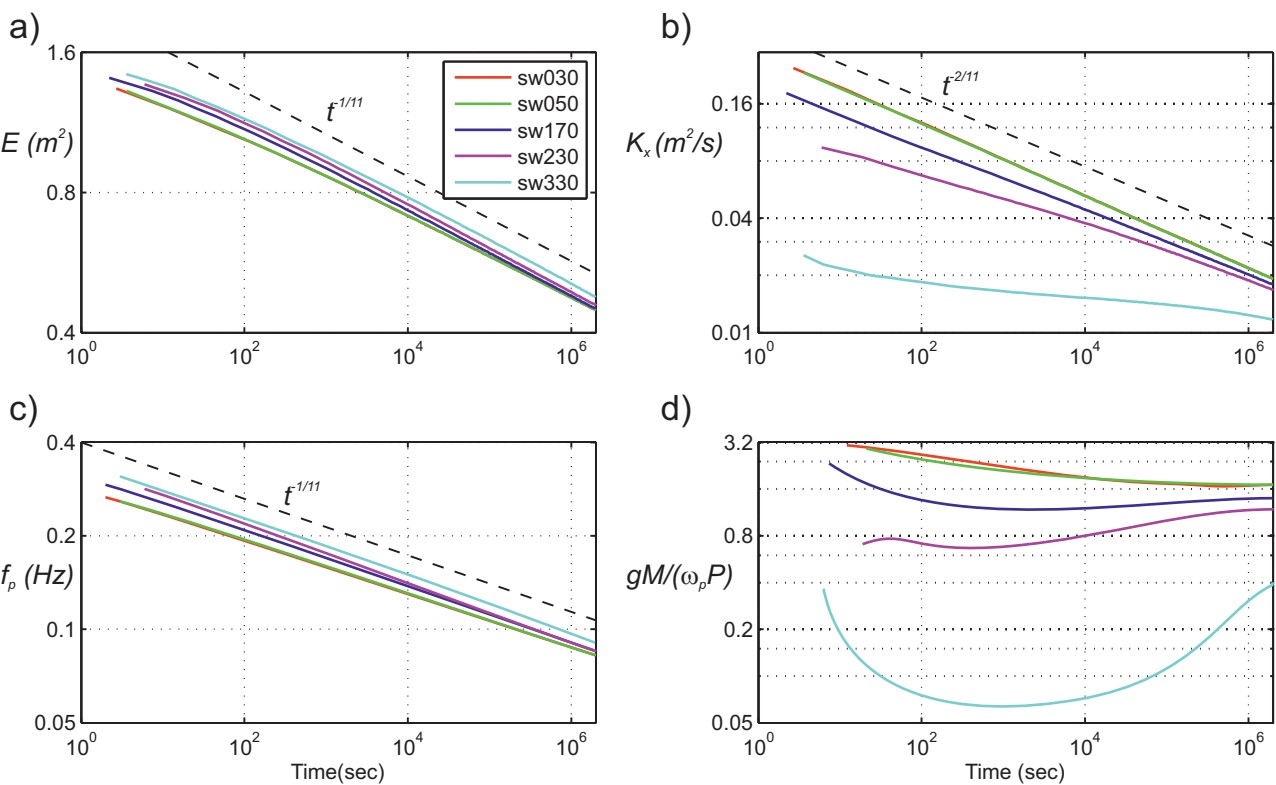

**Figure 2.** Evolution of wave parameters for runs of Table 1 (in legend): *a)* – total energy $E$; *b)* -total wave momentum $K_x$; *c)* – frequency $f_p = \omega_p/(2\pi)$ of the energy spectra peak; *d)* – estimate of parameter of anisotropy $A$ (26). Dashed lines show asymptotic power laws (22,24)

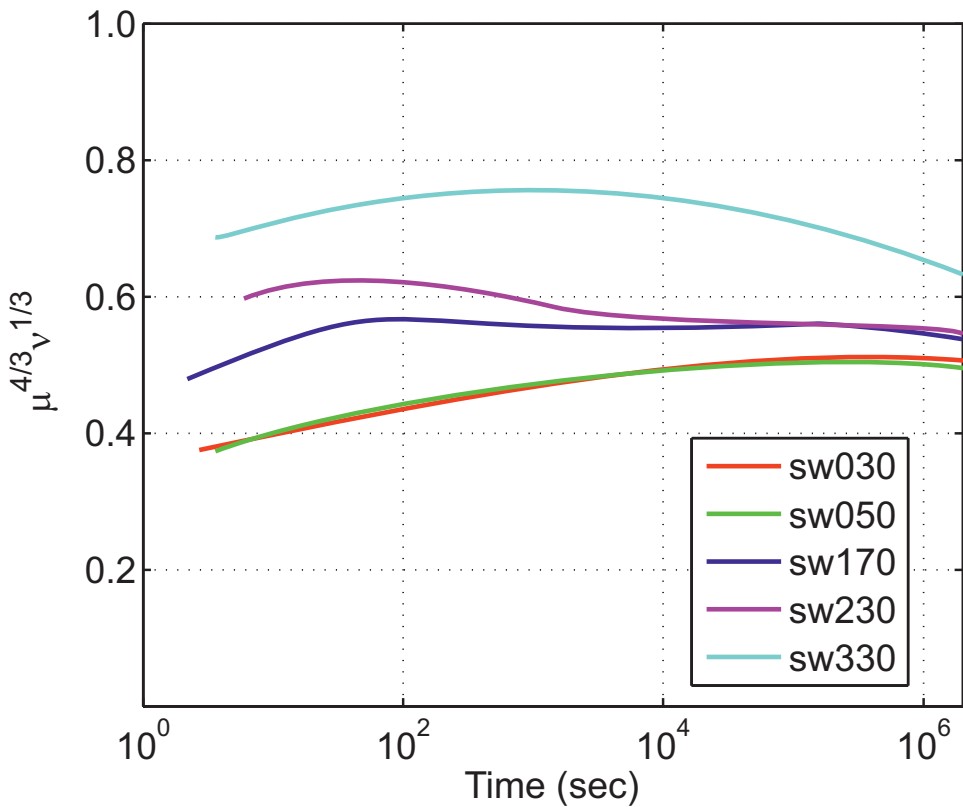

**Figure 3.** Evolution of the left-hand side of the invariant (14) $(\mu^4\nu)^{1/3}$ for runs of Table 1 (in legend).

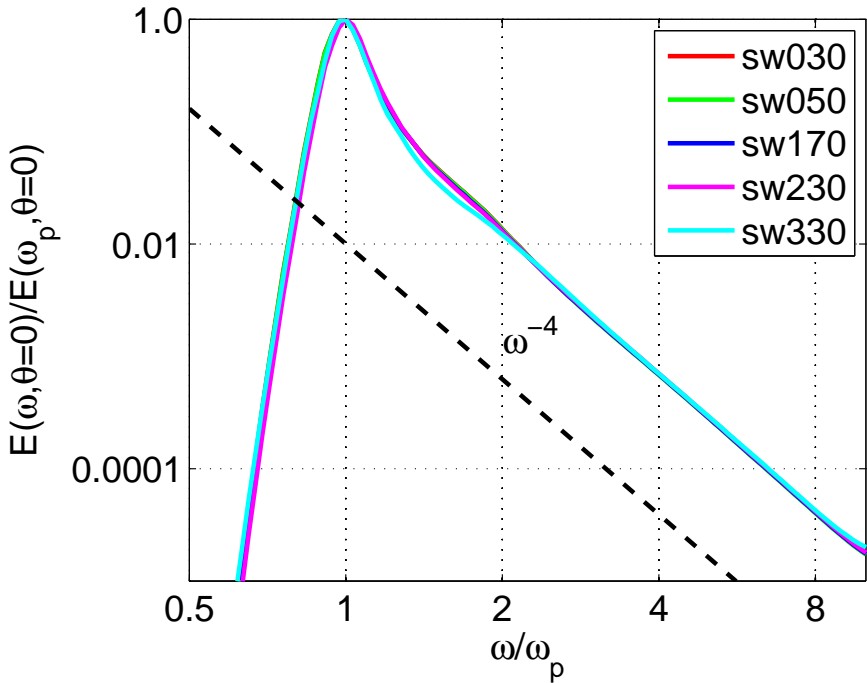

**Figure 4.** Normalized frequency spectra for direction $\theta = 0°$ after 11.5 days (approximately $10^6$ seconds) of swell evolution for runs of Table 1 (see legend).

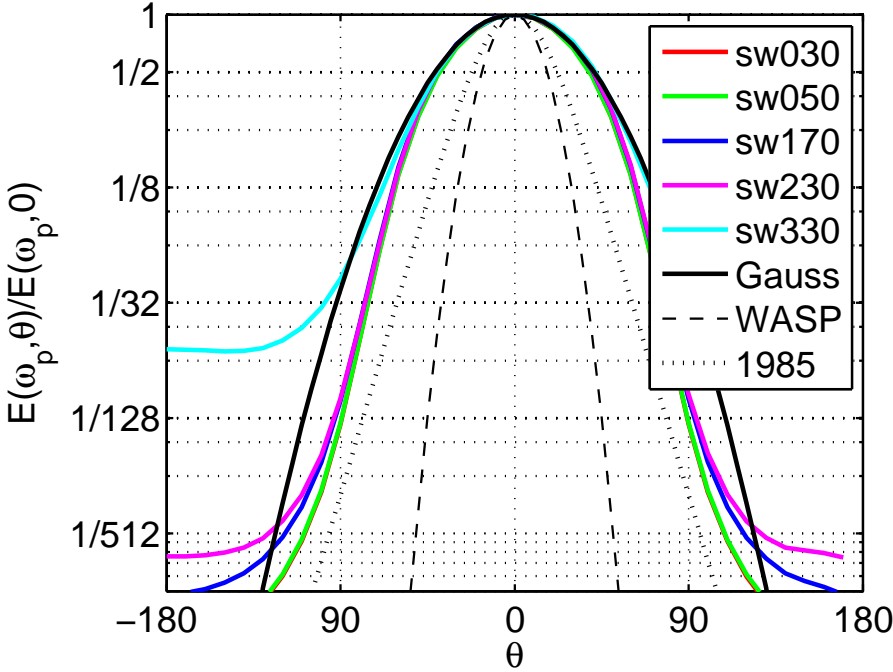

**Figure 5.** Normalized dependence of swell energy spectra on angle at peak frequency $\omega_p$ after 11.5 days (approximately $10^6$ seconds) of swell evolution for runs of Table 1 (see legend). Dashed line – Gaussian distribution (29) with dispersion $\sigma_\Theta = 35°$; dotted – growing sea (30) and eq.9.1-9.2 of Donelan et al. (1985); dashed – wrapped-normal fit of Ewans et al. (2004, Table 11.2, fig.11.8 in ).

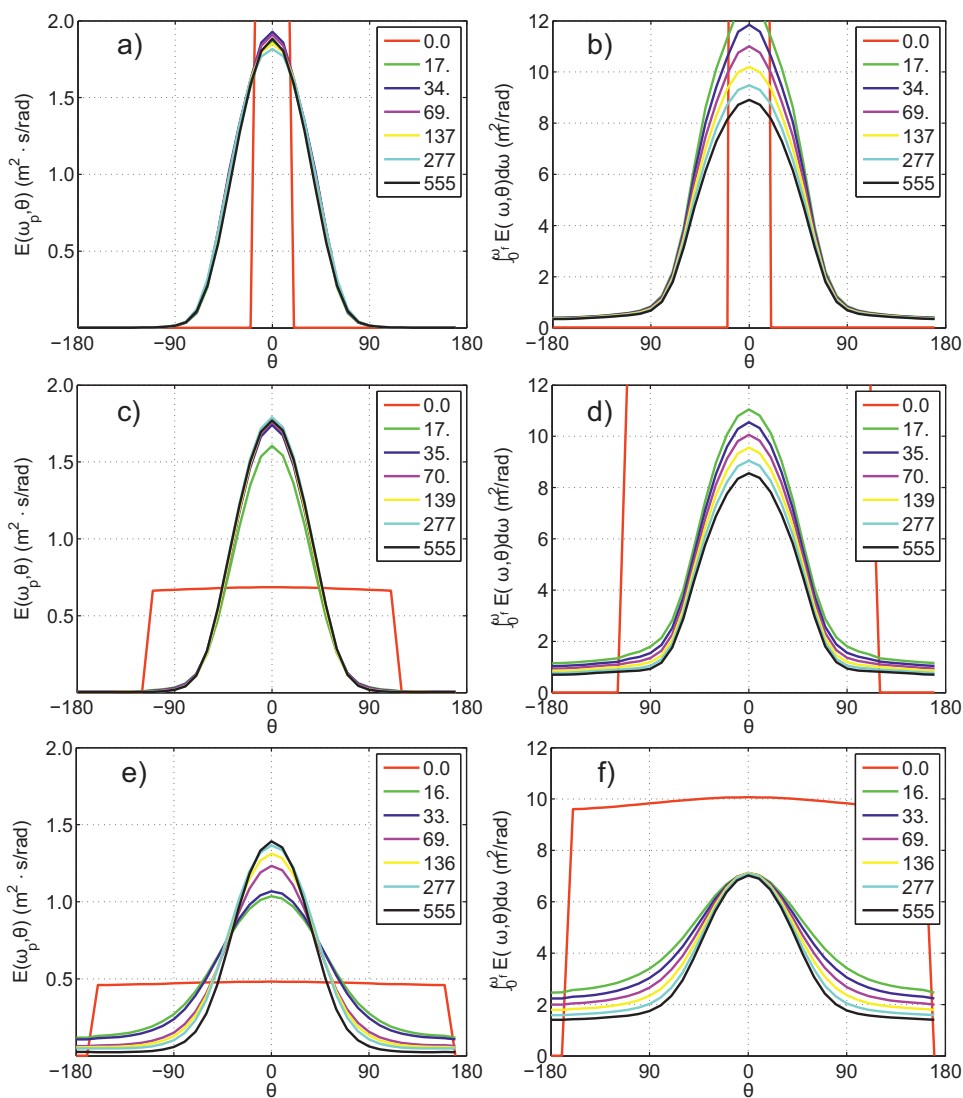

**Figure 6.** Angular spreading of the swell spectra at different times (in hours, see legend). Left column – wave spectra at peak frequency, right – integral of wave spectra in frequency as function of direction. *a,b)* – run sw030 of Table 1 – strong initial anisotropy; *c,d)* – run sw230 – weak anisotropy; *e,f)* – 'almost asitropic' run sw330.

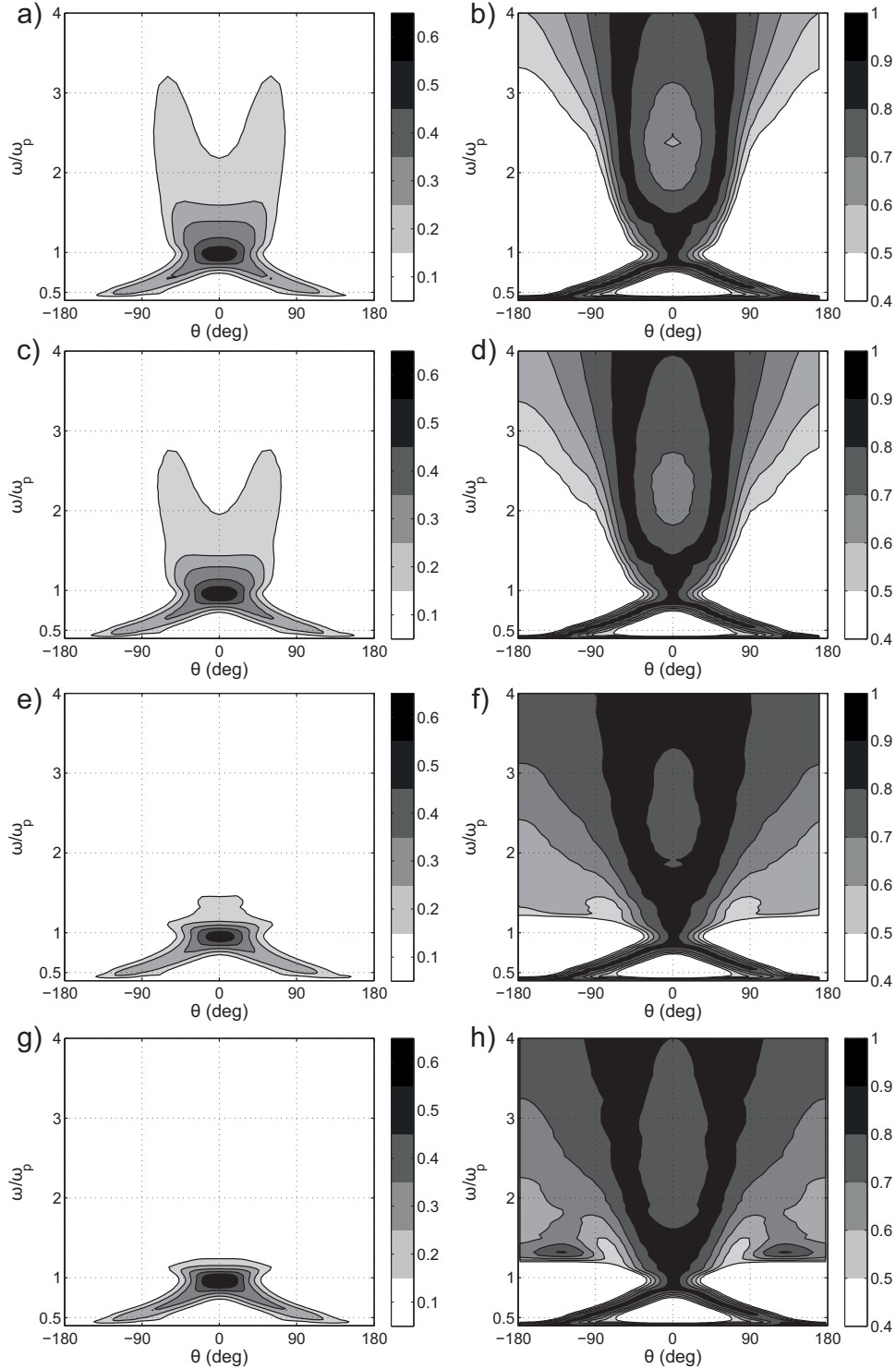

**Figure 7.** Isolines of spreading functions for different runs (see Table 1) *a,b)* – sw030; *c,d)* – sw170; *e,f)* – sw330; *g,h)* – run sw330 with finer resolution in angle $\Delta\theta = 5°$. Left column – definition (32), right – (33).

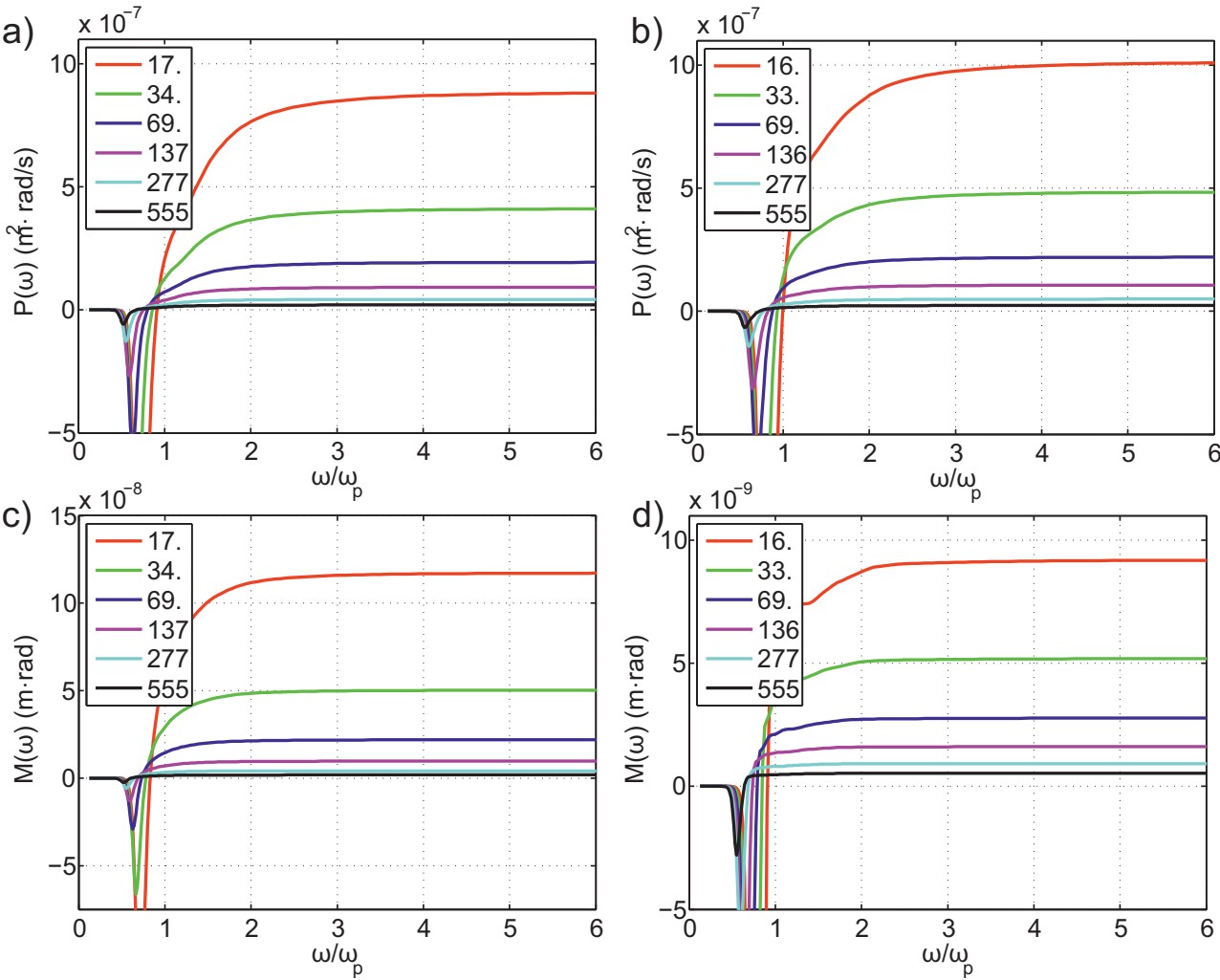

**Figure 8.** Top raw – spectral fluxes of energy for series `sw030` *(a)* and `sw330` *(b)*, bottom raw – spectral fluxes of momentum for series `sw030` *(c)* and `sw330` *(d)* at different times (legend, in hours).

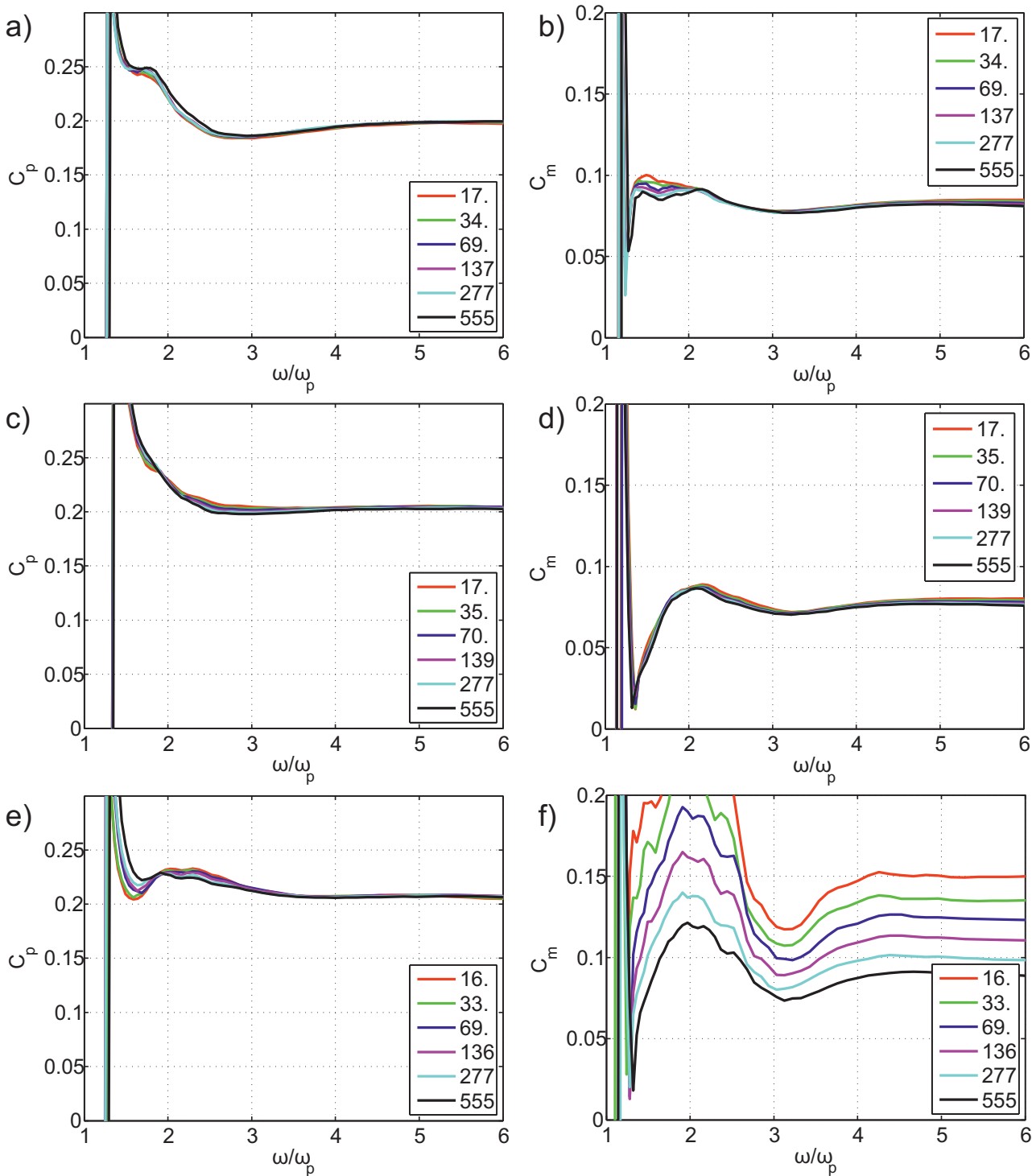

**Figure 9.** Left – estimates of the first Kolmogorov constant $C_p$, right – estimates of the second Kolmogorov constant $C_m$ for the approximate anisotropic KZ solution (6). *a,b)* – run `sw030`; *c,d)* – `sw230`; *e,f)* – `sw330`. Time in hours is given in legend.

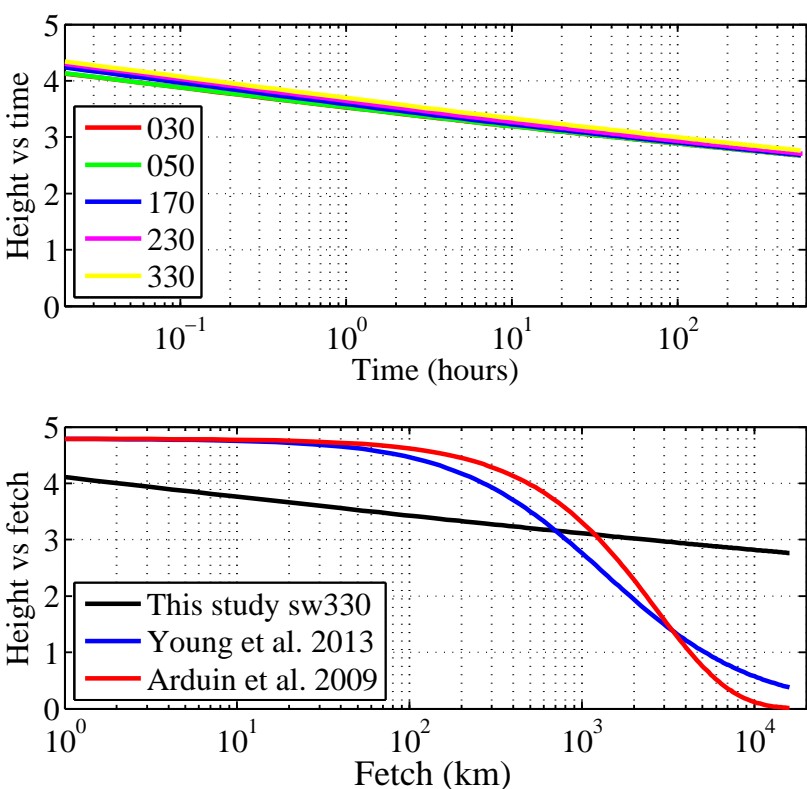

**Figure 10.** Top – dependence of significant wave height $H_s$ on time for cases of Table 1. Bottom – attenuation of swell for models Ardhuin et al. (2009); Young et al. (2013) and one of this paper (see ledend). Results of duration-limited simulations are recasted into dependencies on fetch by simple transformation (36).