# Peer review of "OCEAN SWELL WITHIN THE KINETIC EQUATION FOR WATER WAVES"

_Nonlinear Processes in Geophysics, 2016_

## Referee Comment (RC1) · Anonymous Referee #1 · 8 Dec 2016

Nonlinear Processes Geophysics, NPG-16-

Ocean Swell within the kinetic equation for water waves Zakharov and Badulin

The authors present interesting notions about swell evolution for long durations and it seems worth publishing. Based on my considerations I recommend a major review.

There are 3 important notions that should be discussed or modified in more detail. One concerns the model setup used for the numerical simulations, the second concerns the discussion of the initial swell decay for the near field, and the third concerns the role of weak winds.

A major assumption, of which the consequences are not yet clear, is that swell evolution is treated as duration-limited evolution in an infinitely homogeneous ocean, which effectively reduces the action balance equation to dN/dt=Snl4. This assumption neglects dispersion and spatial divergence of wave energy. This mismatch makes it difficult to compare the results of this study with observations. The consequences of this assumption in relation to the true evolution of swells on the ocean surface need to be clarified

The second point concern the different phases of swell decay in the form of near-field and far-field. The authors argue that there is an initial strong swell decay. The relatively strong decay of the near-field is still hypothetical, as no comparison with field observations could be made. Whether such measurements do not exist, or whether the authors have not searched for such measurements, is unclear. Looking at the results in the Figures 8 and 10 I conclude that the comparison against field data is made on the basis of the sw330 case. This seems a bad choice for 2 reasons. Firstly, sw330 is not comparable to field data in terms of directional spreadings. Secondly, the results in Figure 8 show for the initial phases a strong flux for the higher frequencies, causing this decay. For this case, the spectrum is initially very wide, and the nonlinear interaction try to narrow the spectrum towards an equilibrium situation, meanwhile pumping a lot of energy to the spectral tail.

Thirdly, the role of a weak wind in even strengthening swells is much too hypothetical. Within certain assumptions this may be the result of a theoretical exercise, but I doubt whether unstated assumptions hold. In my feeling, a weak wind will lead to additional energy way beyond the swell peak of the spectrum, effectively changing the shape of the spectrum, causing a mismatch of self-similar spectra. A simple numerical test should be performed to shed more light on this issue.

In the present manuscript a directional spreading of 30° is considered to be very narrow. This seems a proper choice, and the authors may refer to observations where directional spreadings in the order of 10°-15° are described (Olagnon et al., 2013; Ewans 1998).

The English style of writing is good, although at some places small grammatical errors are made. One last round of a native speaker is recommended when this manuscript may reach its final stage. I am very satisfied with the quality and clarity of the figures.

Some detailed remarks, also recommendations for improved readability Page number/Line number

2/2 briefly explain the concept of c-folding 2/5 elaborate on the algebraic law, for which process is such a law made 2/10 I disagree with the generality of the statement that swell is considered a superposition of sinusoidal components without interaction. Maybe in the time of Barber and Ursell (1948), and Snodgrass (1966) and before the time of 3G-wave models. Although I agree that the DIA in the WAM model is not a nice example due to its limitations. 2/14 Briefly explain concept of e-folding 2/15 You may reference to Kantha (2006) here concerning theories about swell decay. 2/21 Which other motions are meant here? 2/33 Add assumption of deep water and also note corresponding period range of 10 s – 16 s 3/9 A useful reference here is Delpey et al., 2009 3/10 Note that wave dispersion and spatial divergence are considered important in ocean scale swell propagation, although for distances over 10.000 km convergence kicks in. 3/13 The swell heightening by a weak background wind is rather speculative, see comments in appropriate section. I would not yet consider this a significant problem from a practical point of view. From a theoretical point of view it is interesting to figure out exactly what is happening. 4/3 The scaling law (2) only works when spectra are self-similar, which may not be the case in nature. 5/7 I would rather drop the very before preliminary. Otherwise, this result is not worth publishing yet. 8/1 The model setup should be specified in more detail. Just referencing to Badulin et al. 200X is insufficient . After some checking it appear that a 1-point model is used to mimic duration limited wave growth, see e.g. Eq.6 in Badulin et al. (2005). This is an important detail, especially since it violates the statement on page/line 3/10. 8/8 10° resolution may be adequate, although no reasoning is shown to back this claim, for the present application where 30° is the smallest directional spreading. In am not convinced whether

this is sufficient for ocean swells in nature, where directional spreading in the range of 10° - 15° are common. For such situations a directional resolution of 5° is usually recommended. 8/10 The equation has some problems. The square 2 is at the wrong location. Further, the variables on each side of the equal sign are inconsistent. I suggest to use N(k,ïĄś) in the left-hand side. The frequencies ïĄů_l and ïĄů_h are not specified. 8/17 Explain concept of hyper-dissipation, just the key notion is sufficient. 8/19 Why mention here the number of 30 runs, whereas the table 1 only contains 5 entries? What happened with the other 25 runs. 9/7 If 11 days is too short, why not extend the simulation longer? On the other hand, the earth's oceans may be too small to see this effect in nature. This poses a conflict, in the applicability of these results. There is only a tendency to approach self-similar solutions. 10/10 Which definition of sigma is used: the linear or the circular definition. Note that the latter is commonly used in wave model to quantify the directional spreading 10/13 Take a look at Ewans (1998) and Olagnon et al. (2013) for realistic estimates of swell widths, these are close to your definition of directional narrowness of ïĄŚ=30°. 11/15 Equation number (31) is missing here. Renumber all follow-up equations 11/18 There are also negative fluxes! 11/26 Why not provide the other estimates for the reader to judge whether the results of this study are consistent? 12/14 I am still surprised by this statement that such attenuation has never been seen in nature. Is it the result of your model setup of using only a 1-point model and only duration limited wave growth? 12/25 I wonder whether the case shown in Figure 10 is properly chosen. Sw330 can hardly be seen as representative for ocean swell in nature. Why not use the case sw030 here to illustrate the point. Now, I am afraid that completely different types of spectra are inter-compared, leading to false interpretation. 13/20 Although the algebra may be trivial, mention the starting point of this exercise 13/32 This may appear an interesting result, but it is only valid within certain assumptions of self-similar spectra. I doubt that this condition holds in case of some wind growth. I expect that some local enhancement of spectral density will appear, which will not cause any effect on the low-frequency part. Having said that, only detailed numerical experiments can shed light on this issue. So, I welcome this

hypothesis, but for now it do not (yet) believe in this consequence. 14/1 I disagree with the choice of the word 'clearly', see my previous comment. It is only an hypothesis within some assumptions. 14/11 Also quantitatively? 14/15 I disagree that this can be used as a benchmark for real ocean swells in view of the limited size of earth's oceans. See comment 9/7. 14/25 I disagree that today's models do not account for this effect. In case of the DIA, the most common method for Snl4, this may be crude or wrong, but it does something. 14/25 I am not convinced that this 'near field' effect has never been observed or noted. It is now too easy stated that this is a problem. Still, it is an interesting notion for further investigations 15/8 This is an interesting statement, but in view of comment 8/1 both dispersion and spatial divergence are important. Only a true 2-d spherical model of swell propagation over the oceans can shed light on this issue. It is disappointing that this notion is not mentioned by the authors. 15/12 No clear recommendations are given for further studies. See also previous point, which is probably one of the most important steps forward. 16/11 This reference cannot be found on the workshop website, only the abstract resides there. 16/32 The journal of Chen et al., 2002 is wrong. Please correct. 19 Table 1 only list 5 of the 30 cases. What are the remaining 25 cases? 20 The initial shape at t=0 does not match with Eq. 23. 20 The unit along the vertical axis is incomplete > m^2/(rad/s) 22 How do you explain the significant mismatch in behavior for case sw330? 24 It is known that Snl4 is weaker in directions than in frequencies to show self-similar behavior. This was for instance noted in the directional response behavior of the spectrum after a change in wind direction. I do not think the 1984 and 1985 are proper examples. See also re-mark 10/30. 25 The scale of the vertical axis is inconsistent with the one in Figure 5. 27 I am surprised that case sw170 is used is as an example. This deviates from other choices. Please comment on or argue this choice. Also, note the small instability for t=1 hour. Also note that also the negative fluxes tend to diminish. Also, argue choice of sw170 for this example. What happens for other choices? In general, the behavior of sw030 or sw050 is much more interesting in relation to real ocean swells. Although, it is of interest that even for initial broad spectra, Snl4 tends to force a uniform shape.

28 Same comment in relation to choice of SW170 29 I am surprised that for this figure sw330 is taken to compare with observations. Why not sw030 or sw050 as that is much closer to field data 29 There is an inconsistency between figure legend and body text concerning reference to Badulin.

References: Barber and Ursell, 1948, The generation and propagation of ocean waves and swell. Proc. Roy Soc., A824. Kantha, 2006, A note on the decay rate of swell, Ocean Modelling, 11. Olganon, M., K. Ewans, G. Forristall, M. Prevosto, 2013, West Africa Swell Spectral Shapes. OMAE2013-11228. Delpey, Ardhuin, Collard, Chapron, 2010, Space-time structure of long ocean swell fields. JGR,115. Ewans, 1998, Observations of the directional distribution of fetch-limited waves. J. Phys. Oceanogr. 28.

---

## Referee Comment (RC2) · Anonymous Referee #2 · 22 Dec 2016

The authors perform long time numerical simulations of swell evolution using the Wave Kinetic Equation (WKE). The authors first show that the Kolmogorov Zakharov analytical solutions of the WKE can be observed numerically; then the analysis is devoted to the study of self-similar evolution of the swell: at short time a fast drop of energy is observed while for larger times nonlinear interactions shape in a quasi-universal way the angular distribution of the spectrum. The results are in general interesting and new. However, before publication I would like the authors to comment on the following points:

1) The simulations are made for a very long time scale; could higher order effect in the kinetic equation take place (e.g. five wave interactions??)

2) Line 9, page 6: while discussing the two-lobe structure of the higher frequency part

of the spectrum, the authors state that the the appearance of such structure is generally discussed as an effect of wind. This is only partially true, indeed, the role of nonlinearity in the formation of the lobes has been already discussed in Toffoli, Alessandro, et al. "Development of a bimodal structure in ocean wave spectra." Journal of Geophysical Research: Oceans 115.C3 (2010).

3) in eq. (16) the letter $\nu$ has already been used for the degree of homogeneity of the wave action.

4) Please, comment more on the fact that the "wave action is the only true integral of the kinetic equation".

5) Please, explain what do the authors mean by "free boundary condition" (line 15 page 8)

6) How much the reduction of the wave energy (H_s) depend on the high frequency cut off in the simulations?

7) English should be improved.

---

## Author Comment (AC1) · 21 Feb 2017

**Answers to referee #1**

Authors are grateful to the referee for attentive reading of the manuscript and valuable comments and suggestions. The authors took all these comments into account when preparing the revised version. Many changes are made in the text, almost all the figures have been re-drawn, additional numerical runs have been carried out as recommended by the referee for longer duration and with higher directional resolution. Ten new references appeared in the paper bibliography. Finally, the paper becomes three pages longer. A native speaker who checked the text has made very few suggestions in English style and grammar.

Our answers follow the reviewer's report (given in bold).

**Major remarks:**

1. **A major assumption, of which the consequences are not yet clear, is that swell evolution is treated as duration-limited evolution in an infinitely homogeneous ocean, which effectively reduces the action balance equation to dN/dt=Snl4. This assumption neglects dispersion and spatial divergence of wave energy. This mismatch makes it difficult to compare the results of this study with observations. The consequences of this assumption in relation to the true evolution of swells on the ocean surface need to be clarified**

   The authors realizes severe limitations of the duration-limited setup in the problem of ocean swell. Nevertheless, even this extremely restrictive model shows quite rich physics: self-similarity of swell evolution, universality of spectral shaping, bi-modality of directional spreading. Limitations of the duration-limited setup are now emphasized in many parts of the text (e.g. 3/17-25 Page/Line). Everywhere in the text we stress robustness of the effects of wave-wave interactions and present prospective plans for more realistic models of swell evolution in time and space where wave dispersion and spatial divergence play important roles. We also note concistency of our results with previous numerical and experimental findings (e.g. Banner & Young, 1994; Ewans *et al.*, 2004);

2. **The second point concern the different phases of swell decay in the form of near- field and far-field. The authors argue that there is an initial strong swell decay. The relatively strong decay of the near-field is still hypothetical, as no comparison with field observations could be made. Whether such measurements do not exist, or whether the authors have**

**not searched for such measurements, is unclear. Looking at the results in the Figures 8 and 10 I conclude that the comparison against field data is made on the basis of the sw330 case. This seems a bad choice for 2 reasons. Firstly, sw330 is not comparable to field data in terms of directional spreadings. Secondly, the results in Figure 8 show for the initial phases a strong flux for the higher frequencies, causing this decay. For this case, the spectrum is initially very wide, and the nonlinear interaction try to narrow the spectrum towards an equilibrium situation, meanwhile pumping a lot of energy to the spectral tail.**

We agree that the near-field behavior of the ocean swell is extremely difficult to explore experimentally. This is why we consider our results on the role of wave-wave interactions in the near field as important. Discussion of directional spreading of swell is now extended. Illustrations are given both for narrow and wide directional distributions (figs.6,7). Swell attenuation in fig.10 is presented for all the runs of Table 1: angular spreading has no essential effect on rate of wave energy leakage. At the same time, initially wide spectra (e.g. run sw330) demonstrate quite strong transformation of angular spreading (fig.7) and essential deviations from the stationary KZ reference in terms of the second Kolmogorov constant $C_m$ (fig.9f);

3. **Thirdly, the role of a weak wind in even strengthening swells is much too hypothetical. Within certain assumptions this may be the result of a theoretical exercise, but I doubt whether unstated assumptions hold. In my feeling, a weak wind will lead to additional energy way beyond the swell peak of the spectrum, effectively changing the shape of the spectrum, causing a mismatch of self-similar spectra. A simple numerical test should be performed to shed more light on this issue. In the present manuscript a directional spreading of $30°$ is considered to be very narrow. This seems a**

**proper choice, and the authors may refer to observations where directional spreadings in the order of $10° − 15°$ are described (Olagnon et al., 2013; Ewans 1998).**

We do not consider the mechanism of wind wave absorption by swell as hypothetical. This effect has been discussed for experimental data (e.g. Pettersson, 2004; Young, 2006) and in numerical simulations of the Hasselmann equation (Badulin *et al.*, 2008). In the updated paper we analyze this physical effect as a competition of two spectral fluxes: direct cascade produced by swell and inverse cascade of wind-driven waves. Wind waves in this scheme are attempting to grow but are just feeding the swell because of relatively fast relaxation to the inherent swell state (see eq.37 for the relaxation rate). The concise estimate (eq.40) looks quite suggestive for possible experimental verification. The authors are grateful to the reviewer for addressing to works on swell evolution (e.g. Ewans *et al.*, 2004) that gave important experimental illustrations of our results.

**Minor remarks** (Page number/Line number):

1. **2/2 briefly explain the concept of e-folding**
   Explained in lines 2/2: 'Their e-folding scale (distance in which an exponentially decaying wave height decreases by a factor of $e$) about $4000$ km is consistent with some today results...';

2. **2/5 elaborate on the algebraic law, for which process is such a law made.**
   Now 2/6. We added comments on the model deficiency. The mentioned model relies upon a number of questionable hypothesis and empirical observations and cannot be incorporated straightforwardly into existent wave models in a mathematically consistent way;

3. **2/10 I disagree with the generality of the statement that swell is considered a superposition of sinusoidal components without interaction. Maybe in the time of Barber and Ursell (1948), and Snodgrass (1966) and before the time of 3G-wave models. Although I agree that the DIA in the WAM model is not a nice example due to its limitations.**

We made the statement less radical (2/14): 'at most' instead of 'generally'. Unfortunately, the simplistic treatment of the swell is dominating today in time of 3G-wave models. We may refer to the feedback of the associate editor of Journal of Geophysical Research (the very first version of our paper has been rejected from JGR as it is mentioned in the submission form of NPG). Prof. Bruno Castelle wrote:

*'Swell is rather unidirectional and monochromatic once it travels outside the storm area, the resonant interactions for such conditions should therefore be negligible, in contrast with your numerical experiments using a 'rectangular' spectral distribution'.*

One of the referees of the JGR continues:

*'As these waves propagate away from the storm generation site, frequency dispersion means that they separate out into almost monochromatic wave trains of the same frequency. These single frequency waves then propagate across oceanic basins and gradually decay.'*

Thus, the hypotheses and the very first physical models of the ocean swell of brilliant papers by Barber and Ursell (1948), Snodgrass et al. (1966) are still alive without critical revision and without attentive reading of important parts of these works (e.g. sect.8 of Snodgrass et al., 1966);

4. **2/14 Briefly explain concept of e-folding**

It is explained above (see 2/2);

5. **2/15 You may reference to Kantha (2006) here concerning theories about swell decay.**
Thank you, it is just to the point (see ref. in 2/5);

6. **2/21 Which other motions are meant here?**
The issue is detailed, a reference is added (2/25);

7. **2/33 Add assumption of deep water and also note corresponding period range of 10 s–16 s**
Thank you, done (3/2);

8. **3/9 A useful reference here is Delpey et al., 2009**
Thank you for the useful link. It is cited now (3/16);

9. **3/10 Note that wave dispersion and spatial divergence are considered important in ocean scale swell propagation, although for distances over 10.000 km convergence kicks in.**
The authors agree. It is noted in the revised text (e.g. Introduction and Discussion);

10. **3/13 The swell heightening by a weak background wind is rather speculative, see comments in appropriate section. I would not yet consider this a significant problem from a practical point of view. From a theoretical point of view it is interesting to figure out exactly what is happening.**
Effects of the swell 'eating' wind-driven waves are described in Young (2006); Kahma & Pettersson (1994) and reproduced numerically in Badulin *et al.* (2008). In this paper we just propose a tentative estimate of conditions when this effect can play. The discussion of this effect is extended, see sect.4.2 ;

Interactive
comment

11. **4/3 The scaling law (2) only works when spectra are self-similar, which may not be the case in nature.**
    It is not correct. The homogeneity property (2) is valid for any function $N(\mathbf{k})$. It is purely mathematical fact that can be checked easily by simple change of variables in the collision integral $S_{nl}$;

12. **5/7 I would rather drop the very before preliminary. Otherwise, this result is not worth publishing yet.**
    You are right, thank you. Fixed in 5/9-10;

13. **8/1 The model setup should be specified in more detail. Just referencing to Badulin et al. 200X is insufficient. After some checking it appear that a 1-point model is used to mimic duration limited wave growth, see e.g. Eq.6 in Badulin et al. (2005). This is an important detail, especially since it violates the statement on page/line 3/10.**
    Description of the model setup is extended (see sect.3.1). We see no contradiction with the statement of 3/10 if we treat 1-point (in the words of the reviewer) and duration-limited setups as synonyms;

14. **8/8 $10°$ resolution may be adequate, although no reasoning is shown to back this claim, for the present application where $30°$ is the smallest directional spreading. In am not convinced whether this is sufficient for ocean swells in nature, where directional spreading in the range of $10° - 15°$ are common. For such situations a directional resolution of $5°$ is usually recommended.**
    Calculations with $5°$ resolution have been carried out for 'the most inconvenient' runs sw030 and sw330 for the duration $2 \cdot 10^6$s. No difference in evolution of integral parameters (energy, momentum, spectral peak period) is found while quantitative difference in angular distributions is visible for frequencies higher than the peak one (fig.7e-h). Comments and new figures are given in the paper

version. Robustness of the two-lobe angular distribution is stressed in sect.3.4. The necessity of higher directional resolution is stressed in final lines of the paper (18/25);

15. **8/10 The equation has some problems. The square 2 is at the wrong location. Further, the variables on each side of the equal sign are inconsistent. I suggest to use $N(\mathbf{k}, \theta)$ in the left-hand side. The frequencies $\omega_l$, $\omega_h$ are not specified.**
Thank you. The typo is corrected. The expression in terms of $\theta$ and $\omega$ for the spatial spectrum looks more transparent (the issue of $N(\mathbf{k}) = \text{const}$). Comments to the eclectic presentation are given to explain our preferences (8/19);

16. **8/17 Explain concept of hyper-dissipation, just the key notion is sufficient.**
We added the comment in sect.3.1 (8/26 and below). In earlier versions of the code (Pushkarev *et al.*, 2003) the hyper-viscosity option has been used to guarantee stability of calculations at high frequencies. Later on it has been realized that calculations can be stable in absence of dissipation (free boundary conditions). The sufficiently strong dissipation does not essentially affect numerical solutions: dissipation is stronger – spectral magnitudes are lower, and the overall effect of the dissipation reaches a sort of saturation. The dissipation effect just absorbs a spectral cascade directed to small (infinitely small) wave scales. Free boundary conditions work in a similar way;

17. **8/19 Why mention here the number of 30 runs, whereas the table 1 only contains 5 entries? What happened with the other 25 runs?**
Initial conditions are now described for all the series after 9/3. We focused on runs of Table 1 that cover the full set of angles (effect of anisotropy is our priority) and have no troubles with possible instability or too slow evolution;

18. **9/7 If 11 days is too short, why not extend the simulation longer? On the other hand, the Earth's oceans may be too small to see this effect in nature.**

**This poses a conflict, in the applicability of these results. There is only a tendency to approach self-similar solutions.**

Calculations for our main series (Table 1) are extended to $2 \cdot 10^6$ s to better specify tendency of wave parameters (height, period) and spectral shapes to a self-similar behavior and to specify 'pure effect' of nonlinear transfer due to four-wave interactions. It appears again 'too short'. Anyway, the tendency to self-similarity is better than tendency to nowhere. 'The effect in nature' requires an advanced setup with wave dispersion and spatial divergence/convergence taken into account as the reviewer himself stressed;

19. **10/10 Which definition of sigma is used: the linear or the circular definition. Note that the latter is commonly used in wave model to quantify the directional spreading**

    Linear definition (in degrees) of $\sigma$ and $\theta$ is used everywhere in the text and in figures. Hope, it makes no problem for the paper potential readers;

20. **10/13 Take a look at Ewans (1998) and Olagnon et al. (2013) for realistic estimates of swell widths, these are close to your definition of directional narrowness of $\Theta = 30°$.**

    Thank you for this reminder. We had the authentic report of Ewans *et al.* (2004) and now use it in the updated text. This work give extremely wide range of estimates of directional spreading. We knew about this report when preparing the first version of the paper but it seemed too radical in following linear model of swell propagation. Now this and other papers (e.g. Ewans, 2001) are cited in the context of angular spreading of swell (sects.3.3,3.4);

21. **11/15 Equation number (31) is missing here. Renumber all follow-up equations**

    There are no references to this equation in the text below. We leave the equation unnumbered;

22. **11/18 There are also negative fluxes!**
You are right. We added 'negative' and 'positive' in the previous paragraph when discussing the hybrid nature of swell solutions (13/9, 13/10);

23. **11/26 Why not provide the other estimates for the reader to judge whether the results of this study are consistent?**
The values are provided (14/1-6), a reference (Deike *et al.*, 2014) to an experimental estimate of $C_p$ is added;

24. **12/14 (Likely, 14/24) I am still surprised by this statement that such attenuation has never been seen in nature. Is it the result of your model setup of using only a 1-point model and only duration limited wave growth?**
The effect of attenuation of swell has never been discussed as one observed in nature. Other 'visible' mechanisms of swell decay like spatial dispersion or dissipation are in the focus of swell studies. Moreover, the fact itself of non-conservation of wave energy and momentum is not accepted by majority of researchers (Janssen, 2004, p.182, comments to eq.4.20 or p.137, sect. *Conservation laws* in Komen *et al.* (1995)),

25. **12/25 I wonder whether the case shown in Figure 10 is properly chosen. Sw330 can hardly be seen as representative for ocean swell in nature. Why not use the case sw030 here to illustrate the point. Now, I am afraid that completely different types of spectra are inter-compared, leading to false interpretation.**
Figure 10 is re-drawn. Upper panel shows all runs of the series with no essential quantitative difference. Thus, our choice representative. See also comments to page 29 below;

26. **13/20 Although the algebra may be trivial, mention the starting point of this exercise**
It is given in more details in sect.4.2 now;

NPGD

27. **13/32 This may appear an interesting result, but it is only valid within certain assumptions of self-similar spectra. I doubt that this condition holds in case of some wind growth. I expect that some local enhancement of spectral density will appear, which will not cause any effect on the low-frequency part. Having said that, only detailed numerical experiments can shed light on this issue. So, I welcome this hypothesis, but for now it do not (yet) believe in this consequence.**
The effect is seen fairly well in previous numerical experiments (Badulin *et al.*, 2008). We also have new results on this effect and hope to publish them soon;

28. **14/1 I disagree with the choice of the word 'clearly', see my previous comment. It is only an hypothesis within some assumptions.**
Thank you. We deleted it (17/5);

29. **14/11 Also quantitatively?**
Thank you. Now *'quantitatively and even qualitatively'* (17/15);

30. **14/15 I disagree that this can be used as a benchmark for real ocean swells in view of the limited size of earth's oceans. See comment 9/7.**
Thank you. Now *'features KZ solutions can be used as a reference'*. . . (17/29);

31. **14/25 I disagree that today's models do not account for this effect. In case of the DIA, the most common method for $S_{nl4}$, this may be crude or wrong, but it does something.**
Thank you. Now we say: *'This mechanism is beyond the today models of sea swell. . . '* (17/31). The problem can be addressed to the DIA, first, to uncover whether the models are accounting for this effect;

32. **14/25 I am not convinced that this 'near field' effect has never been observed or noted. It is now too easy stated that this is a problem. Still, it is an interesting notion for further investigations**

We did not say 'never been observed and noted'. The today studies of swell from space do avoid discussion the near field effects and, thus, skip an essential physics of sea wave dynamics. The text is modified (bottom of p.17, top p.18);

33. **15/8 This is an interesting statement, but in view of comment 8/1 both dispersion and spatial divergence are important. Only a true 2-d spherical model of swell propagation over the oceans can shed light on this issue. It is disappointing that this notion is not mentioned by the authors.**
Ok, we turn our cards over. Perspectives of the study are given in more details now (18/19 and below);

34. **15/12 No clear recommendations are given for further studies. See also previous point, which is probably one of the most important steps forward.**
Thank you. Corrected, see previous note;

35. **16/11 This reference cannot be found on the workshop website, only the abstract resides there.**
It is a pity. Reference to ResearchGate source of the paper is added. Similarly, the conference paper of Lavrenov *et al.* (2002) is put into supplement of the ResearchGate web-page of Badulin *et al.* (2002) and the corresponding reference is given. Unfortunately, Prof. Igor Lavrenov deceased in 2009 and its paper resides now at this web-page;

36. **16/32 The journal of Chen et al., 2002 is wrong. Please correct. Journal of Atmospheric and Oceanic Technology**
Thank you. Fixed;

37. **19 Table 1 only list 5 of the 30 cases. What are the remaining 25 cases?**
Parameters of simulations are described in more details in sect.3.1;

38. **20 The initial shape at $t = 0$ does not match with Eq. 23.**
We see no problem. Eq.23 (eq.25 now) gives spectral density of wave action

$N(\mathbf{k})$ while Fig.1 shows evolution of energy frequency spectrum

$$E(\omega) = \int_{-\pi}^{\pi} \frac{2\omega^4 N(\mathbf{k}(\omega, \theta))}{g^2} d\theta$$

(see for refs. Badulin *et al.*, 2005, unnumbered equations after eq.30);

39. **20 The unit along the vertical axis is incomplete m$^2$/(rad/s)**
Thank you. Corrected for two times longer evolution;

40. **22 How do you explain the significant mismatch in behavior for case sw330?**
Calculations are continued up to $2 \cdot 10^6$s, Figs.2,3 are redrawn. The explanation can be found in sect.3.2-3.5. The case is 'too isotropic' and non-self-similar background corrupts a bit the simple asymptotics;

41. **24 It is known that $S_{nl4}$ is weaker in directions than in frequencies to show self-similar behavior. This was for instance noted in the directional response behavior of the spectrum after a change in wind direction. I do not think the 1984 and 1985 are proper examples. See also remark 10/30.**
We leave 1985 and added WASP from Ewans *et al.* (2004). Weakness of $S_{nl4}$ in direction is misleading. The relaxation rate depends on magnitude of excursion. This is what we see in fig.6 for sw330. See also 10/30 – speculations on different scales of evolution due to wave-wave interactions;

42. **25 The scale of the vertical axis is inconsistent with the one in Figure 5.**
You are right. In fig.5 normalized (by value at $\theta = 0$) values for different runs are shown while in fig.6 we give absolute values at different times for the same run in order to demonstrate the phenomenon of relaxation to a universal (our hypothesis) angular distribution;

43. **27 I am surprised that case sw170 is used is as an example. This deviates from other choices. Please comment on or argue this choice. Also, note the small instability for $t = 1$ hour. Also note that also the negative fluxes tend to diminish. Also, argue choice of sw170 for this example. What happens for other choices? In general, the behavior of sw030 or sw050 is much more interesting in relation to real ocean swells. Although, it is of interest that even for initial broad spectra, $S_{nl4}$ tends to force a uniform shape.**
    This figure is re-drawn. Results are shown for two extreme cases sw030 and sw330;

    Sorry, we do not see any instability for red curves $t = 1$hr.

    We answered the question on negative fluxes (hybrid nature of swell evolution) in the note 11/18. Negative fluxes follow the same tendency as positive fluxes when solutions are tending to self-similar behavior. We see no reason to emphasize this point here.

    Thanks for your last phrase of this note. You stressed the very important finding of our work: $S_{nl4}$ provides a uniform (we say universal) shapes of swell irrespectively to initial spectral distribution;

44. **28 Same comment in relation to choice of SW170**
    Three cases are shown now in fig.9. The only outlier is sw330 for the second Kolmogorov constant $C_m$;

45. **29 I am surprised that for this figure sw330 is taken to compare with observations. Why not sw030 or sw050 as that is much closer to field data**
    Re-drawn. All cases are shown. Time and coordinate axes are logarithmic now to see the 'near-field' better;

46. **29 There is an inconsistency between figure legend and body text concerning reference to Badulin.**

Thank you. The figure is re-drawn. Time and fetch axes are log-spaced now in order to demonstrate strong drop of wave heights in near zone (less than 1000 km). Curves are given for all runs of the series and show quite close behavior for different initial distributions.

**References**

BADULIN, S. I., KOROTKEVICH, A. O., RESIO, D. & ZAKHAROV, V. E. 2008 Wave-wave interactions in wind-driven mixed seas. In *Proceedings of the Rogue waves 2008 Workshop*, pp. 77–85. IFREMER, Brest, France.

BADULIN, S. I., PUSHKAREV, A. N., RESIO, D. & ZAKHAROV, V. E. 2002 Direct and inverse cascade of energy, momentum and wave action in wind-driven sea. In *7th International workshop on wave hindcasting and forecasting*, pp. 92–103. Banff, October 2002, available at `https://www.researchgate.net/publication/253354120_Direct_and_inverse_cascades_of_energy_momentum_and_wave_action_in_spectra_of_wind-driven_waves`.

BADULIN, S. I., PUSHKAREV, A. N., RESIO, D. & ZAKHAROV, V. E. 2005 Self-similarity of wind-driven seas. *Nonl. Proc. Geophys.* **12**, 891–946.

BANNER, M. L. & YOUNG, I. R. 1994 Modeling spectral dissipation in the evolution of wind waves. part i: Assessment of existing model performance. *J. Phys. Oceanogr.* **24** (7), 1550–1571.

DEIKE, L., MIQUEL, B., GUTIÉRREZ, P., JAMIN, T., SEMIN, B., BERHANU1, M., FALCON, E. & BONNEFOY, F. 2014 Role of the basin boundary conditions in gravity wave turbulence. *J. Fluid Mech.* **781**, 196–225.

EWANS, KEVIN, FORRISTALL, GEORGE Z., PREVOSTO, MICHEL OLAGNON MARC & ISEGHEM, SYLVIE VAN 2004 WASP West Africa Swell Project. Final report. Ifremer - Centre de Brest, Shell International Exploration and Production, B.V.

EWANS, K. C. 2001 Directional spreading in ocean swell. In *The Fourth International Symposium on Ocean Wave Measurement and Analysis, ASCE, San Francisco*.

JANSSEN, P. A. E. M. 2004 *The Interaction of Ocean Waves and Wind*. Cambridge Univ. Press, New York, 300 pp.

[Figure]

KAHMA, K. K. & PETTERSSON, H. 1994 Wave growth in a narrow fetch geometry. *Global Atmos. Ocean Syst.* **2**, 253–263.

KOMEN, G. J., CAVALERI, L., DONELAN, M., HASSELMANN, K., HASSELMANN, S. & JANSSEN, P. A. E. M. 1995 *Dynamics and Modelling of Ocean Waves*. Cambridge University Press.

LAVRENOV, I., RESIO, D. & ZAKHAROV, V. 2002 Numerical simulation of weak turbulent Kolmogorov spectrum in water surface waves. In *7th International workshop on wave hindcasting and forecasting*, pp. 104–116. Banff, October 2002, available at `https://www.researchgate.net/publication/312210314_Lavr_7th _Workshop`.

PETTERSSON, HEIDI 2004 Wave growth in a narrow bay. PhD thesis, University of Helsinki, [ISBN 951-53-2589-7 (Paperback) ISBN 952-10-1767-8 (PDF)].

PUSHKAREV, A. N., RESIO, D. & ZAKHAROV, V. E. 2003 Weak turbulent theory of the wind-generated gravity sea waves. *Phys. D: Nonlin. Phenom.* **184**, 29–63.

YOUNG, I. R. 2006 Directional spectra of hurricane wind waves. *J. Geophys. Res.* **111**, doi:10.1029/2006JC003540.

---

## Author Comment (AC2) · 21 Feb 2017

**Answers to referee #2**

The authors appreciate efforts of the reviewer and his/her valuable comments. The paper is significantly updated: almost all the figures have been re-drawn, additional numerical runs have been carried out as recommended by one of referee for longer duration and with higher directional resolution. Ten new references appeared in the paper bibliography. A native speaker who checked the text has made very few suggestions in English style and grammar. Our answers follow the reviewer's report.

1. **The simulations are made for a very long time scale; could higher order effect in the kinetic equation take place (e.g. five wave interactions?)**

We do not discuss the effect of five-wave (and higher-order) interactions intentionally by a number of reasons.

First, the solution itself of the four-wave (Hasselmann) kinetic equation for long time is a real computational problem. The five-wave extension of the equation is well-known (see sect.5 and eqs. 5.1, 5.2 Krasitskii, 1994) but the authors are not aware of attempts to solve it numerically.

Secondly, the passage to the five-wave kinetic equation is 'of principal significance' in the words of Krasitskii (1994). The account for five-wave interactions is violating the wave action conservation law (wave energy and momentum remain to be formal integrals of the extension) and, thus, makes the theoretical concept of the Kolmogorov-Zakharov cascading and power-law Kolmogorov's spectra inapplicable. The authors set a high value on the theoretical background in this paper;

Finally, our principal goal was to stay within the today concept of wind wave and swell prediction where the four-wave Hasselmann equation plays a key role. Tentative estimates of the effect of five-wave interactions for low steepness swell ($\mu \simeq 0.03$) offer prospects of their rather small effect. Quite interesting issue of wave field short-crestedness at long times (e.g. Badulin *et al.*, 1996) is, evidently, beyond of the paper goals and the statistical theory of sea waves;

2. **Line 9, page 6: while discussing the two-lobe structure of the higher frequency part of the spectrum, the authors state that the appearance of such structure is generally discussed as an effect of wind. This is only partially true, indeed, the role of nonlinearity in the formation of the lobes has been already discussed in Toffoli, Alessandro, et al. "Development of a bimodal structure in ocean wave spectra." Journal of Geophysical Research: Oceans 115.C3 (2010).**
Thank you. This note is extended by references to Pushkarev *et al.* (2003) and Toffoli *et al.* (2010). Note, that the latter paper presents results of simulations for

rather short durations of very few hundreds peak periods, i.e. about one hour only for our swell parameters. An extensive discussion of the spectra bi-modality is given in sect.3.4 with references to experimental (Ewans, 2001; Ewans *et al.*, 2004) and numerical works (Banner & Young, 1994; Young *et al.*, 1995);

3. **in eq. (16) the letter $\nu$ has already been used for the degree of homogeneity of the wave action.**
Thank you. Symbol $\nu$ in (2) is changed to $\upsilon$ now;

4. **Please, comment more on the fact that the 'wave action is the only true integral of the kinetic equation'.**
Comments are given in the cited papers (Zakharov *et al.*, 1992; Pushkarev *et al.*, 2003);

5. **Please, explain what do the authors mean by 'free boundary condition' (line 15 page 8)**
A short comment is added (now line 25, p.8): 'Free boundary conditions were applied at the high-frequency end of the domain of calculations: generally, short-term oscillations of the spectrum tail do not lead to instability, i.e. the resulting solutions can be regarded as ones corresponding to condition of decay at infinitely small scales $(N() \to 0$ when $|| \to \infty)$.'

6. **How much the reduction of the wave energy ($H_s$) depend on the high frequency cut off in the simulations?**
We did not find difference when reduced number of frequency grid points from 128 to 112. This is mentioned in the updated text (line 1, p.9)

7. **English should be improved.**
Thank you. We did our best to make the paper readable

**References**

BADULIN, S. I., SHRIRA, V. I. & KHARIF, C. 1996 A model of water wave 'horse-shoe' patterns. *J. Fluid Mech.* **318**, 375–405.

BANNER, M. L. & YOUNG, I. R. 1994 Modeling spectral dissipation in the evolution of wind waves. part i: Assessment of existing model performance. *J. Phys. Oceanogr.* **24** (7), 1550–1571.

EWANS, KEVIN, FORRISTALL, GEORGE Z., PREVOSTO, MICHEL OLAGNON MARC & ISEGHEM, SYLVIE VAN 2004 WASP West Africa Swell Project. Final report. Ifremer - Centre de Brest, Shell International Exploration and Production, B.V.

EWANS, K. C. 2001 Directional spreading in ocean swell. In *The Fourth International Symposium on Ocean Wave Measurement and Analysis, ASCE, San Francisco*.

KRASITSKII, V. P. 1994 On reduced Hamiltonian equations in the nonlinear theory of water surface waves. *J. Fluid Mech.* **272**, 1–20.

PUSHKAREV, A. N., RESIO, D. & ZAKHAROV, V. E. 2003 Weak turbulent theory of the wind-generated gravity sea waves. *Phys. D: Nonlin. Phenom.* **184**, 29–63.

TOFFOLI, A., ONORATO, M., BITNER-GREGERSEN, E. M. & MONBALIU, J. 2010 Development of a bimodal structure in ocean wave spectra. *J. Geophys. Res.* **115** (C03006).

YOUNG, I. R., VERHAGEN, L. A. & BANNER, M. L. 1995 A note on the bimodal directional spreading of fetch-limited wind waves. *J. Geophys. Res.* pp. 773–778.

ZAKHAROV, V. E., LVOV, V. S. & FALKOVICH, G. 1992 *Kolmogorov spectra of turbulence. Part I*. Springer, Berlin.

---

## Referee Report (RR1)

**Referee's report on the revised manuscript "Ocean swell within the kinetic equation for water waves" by S. Badulin and V. E. Zakharov**

The authors consider long-term evolution of swell within the Hasselmann kinetic equation. The major challenge faced by the authors was that the Hasselmann equation is derived under the assumption of spatial homogeneity, although the effects of spatial divergence are known to play an important role for the swell evolution. At the same time, wave-wave interactions, which are modelled by the equation, are of primary importance for the swell evolution as well, and the authors had to find a rather difficult balance between the features of the evolution they could and could not consider in the study. In general, I think that in the revised version of the manuscript this balance was found succesfully. The paper contains interesting results and can be published. A few minor points are listed below.

1. Page 6/5: *"This implies that the only one (or very few) of an infinite series of eigenfunctions... contributes to wave spectra evolution... This treatment of the heavily nonlinear boundary problem in terms of a composition of eigenfunctions is possible in this case as demonstrated by Zakharov and Pushkarev (1999)"*. I think a slightly more detailed explanation is needed here. I guess the authors mean the eigenfunctions of the diffusion problem that can be derived from the Hasselmann equation, but it is not clear which boundary problem they are referring to, why it is "heavily nonlinear", and Zakharov & Pushkarev (1999), while discussing the diffusion approximation, make no mention of eigenfunctions.

2. Page 6/10: *"Their appearance within the kinetic equation approach is generally associated with wind generation..."*. This expression (there are quite a few such cases in the text) is confusing; apparently, the authors mean "generation by wind". Also, I'm not sure about the relevance of both Bottema & van Vledder references here. The title of Pushkarev et al. (2003) reference is not quite correct.

3. Page 6/20: *"This invariance does not suppose a point-by-point coincidence of properly normalized spectral shapes"*. If it does not require point-by-point similarity, then in what sense we can speak about the invariance? Below, the authors go on to discuss only integral parameters of spectra, so may be not clear to the reader why the invariance was mentioned.

4. Page 9/25: what is the initial steepness? For the final values $H_s = 2.8m$, $T_p = 11.4s$ I get the significant wave steepness $\frac{1}{2}H_s k_p = 0.043$, which corresponds to $\mu = 0.022$ in full agreement with the authors (since $\mu$ in Eq.(15) is half the significant steepness). But for the initial $H_s = 4.8m$ and $T_p = 3s$ I cannot get anything like $\mu = 0.15$ as stated by the authors (which is, by the way, quite high already). Instead, I obtain a value beyond all physically reasonable limits. It would not be an issue for long-term simulations, since the steepness in simulations quickly drops anyway, and the exact form of the initial conditions is irrelevant. However, since the authors put a lot of emphasis

on the near-field attenuation in the manuscript, one may think that the fast initial attenuation may be due to unphysical initial conditions.

5. In figure 2, the labelling of panels is missing, and the notation of wave momentum does not correspond to the caption.

6. Page 10/25: *"This relaxation generally occurs at essentially shorter scales than ones of wind pumping and wave dissipation"*. Apparently, the authors mean "shorter time scales" here.

---

## Author Response (ED1)

Dear Editor,

Please, find the revised version of the manuscript

Ocean swell within the kinetic equation

by Sergei Badulin & Vladimir Zakharov

Minor changes are made as suggested by reviewers. The answers are given in the attached letters. In the marked-up version of the manuscript corrections for the Review#1 are marked in red, for the Review#2 – in blue.

The authors appreciate your cooperation and very helpful suggestions of the reviewers

Yours sincerely,

Sergei Badulin & Vladimir Zakharov

**Answers to referee #1**

Dear Dr. van Vledder,

The authors are immensely grateful for your careful and meticulous reading of the paper and your detailed reports. Your big work really helped us to get wider vision of the problem of sea swell. We hope it makes our results closer to the today needs of sea wave studies. We agree with your general remarks on the outstanding works Munk *et al.* (1963); Snodgrass *et al.* (1966) and, following your suggestions, added more comments to these works in the final version. Very likely these comments do not render all the importance and richness of these papers. Further works will compensate this deficiency.

Thank you for the reference to the extremely instructive documentary. We believe that the useful *Youtube* reference (Page/Line - 2/1) will be acceptable for the scientific publication.

**Minor remarks** (Page number/Line number):

1. **2/9 rephrase sentence. Wording a bit strange**
   Now 2/13. Thank you, changed. *'It makes problems for developing a consistent asymptotic approach within the today statistical (and even dynamical) theories of sea waves where the leading nonlinear term is cubic.'*

2. **2/11 some confusion may arise between the (theoretical) models and the simulation (models) inline 2/13. Both are models but they differ.**
   2/15-18.Now: *'It should be stressed that a number of theoretical and numerical models including those mentioned above treats swell as a quasi-monochromatic wave and, thus, ignores nonlinear interactions of the swell harmonics themselves and the swell coupling with locally generated wind waves. The latter effect can be essential as observations and simulations clearly show (e.g. Kahma & Pettersson, 1994; Pettersson, 2004; Young, 2006; Badulin et al., 2008, and refs. therein).'*

3. **5/10 is the mentioned special study this manuscript or something else. Please be clear.**
   5/13. *'is a subject of further studies'.*

4. **8/23-28 Some additional information should be added to avoid a possible confusion in this block of text. One the one hand, it**

is stated that (explicit) dissipation is absent in the simulation. One the other hand the phrase very strong dissipation is mentioned. Moreover, the drop in $H_s$ is later attributed to leakage (see 17/30). As in this 1-point model no spatial spreading occurs, the initial drop in $H_s$ should be more clearly attributed to a specific effect.

Thank you. It is a problem to explain how a real dissipation substitutes the 'conservative dissipation'. We propose the following explanation in 9/1:

*'Calculations with a hyper-dissipation (e.g. Pushkarev et al., 2003) or a diagnostic tail at the high-frequency range of the spectrum (Gagnaire-Renou et al., 2010) do not affect results quantitatively compared to our simulations without any dissipation. Thus, these 'non-conservative' options can mimic successfully the effect of energy leakage at $|\mathbf{k}| \to \infty$ in our formally non-dissipative problem.'*

5. **17/31 Again some clarification is needed about the concept model. Do the authors expect that an accurate estimation of $S_{nl4}$ in a third-generation model can reproduce these features of swell, and also much better than the DIA?**

The authors think that a special study should be carried out to understand what 'an accurate estimation of $S_{nl4}$' means. In fact, an accurate estimate of the spectral flux rather than a point-by-point accuracy of the term $S_{nl4}$ is required for the effect. We added a short comment in 18/4. *'The today models of sea swell are unlikely to account for this effect. Possible problems of the models are sketched in sect.3.1 when different options of simulation of the 'conservative dissipation' are discussed. All these options require sufficiently large high-frequency range where the short-term oscillations in absence of dissipation or hyper-dissipation can mimic the energy leakage at $|\mathbf{k}| \to \infty$'.*

The authors agree. *'At shorter time scales'.*

**References**

[revised manuscript text omitted]

---

## Author Response (AR3)

Dear Editor,

Please, find the revised version of the manuscript

Ocean swell within the kinetic equation

by Sergei Badulin & Vladimir Zakharov

Minor changes following your and reviewers' suggestions are marked by colors (red - rev.1, blue -rev.2, green - Editor). The answers to reviewers have been given in the previous letters. Technical corrections proposed by Editor are now accepted, minor changes are made in the text. Three new references appeared in the manuscript: Tsimring (1986), Annenkov and Shrira (2006), Onorato et al. (2002).

The authors appreciate your helpful cooperation

Yours sincerely,

Sergei Badulin & Vladimir Zakharov

[revised manuscript text omitted]